# Improving and Accelerating Offline RL in Large Discrete Action Spaces with Structured Policy Initialization

**Matthew Landers**[1]*, **Taylor W. Killian**[2], **Thomas Hartvigsen**[1], **Afsaneh Doryab**[1]
[1]University of Virginia, [2]MBZUAI

## Abstract

Reinforcement learning in discrete combinatorial action spaces requires searching over exponentially many joint actions to simultaneously select multiple sub-actions that form coherent combinations. Existing approaches either simplify policy learning by assuming independence across sub-actions, which often yields incoherent or invalid actions, or attempt to learn action structure and control jointly, which is slow and unstable. We introduce Structured Policy Initialization (SPIN), a two-stage framework that first pre-trains an Action Structure Model (ASM) to capture the manifold of valid actions, then freezes this representation and trains lightweight policy heads for control. On challenging discrete DM Control benchmarks, SPIN improves average return by up to $39\%$ over the state of the art while reducing time to convergence by up to $12.8\times$[1].

## 1 Introduction

Many real-world problems require decision-making in high-dimensional discrete action spaces, including applications in healthcare (Liu et al., 2020), robotic assembly (Driess et al., 2020), recommender systems (Zhao et al., 2018), and ride-sharing (Lin et al., 2018). In such domains, online exploration can be costly or unsafe, making offline reinforcement learning (RL) (Lange et al., 2012; Levine et al., 2020) an appealing framework. Standard offline RL methods (Fujimoto et al., 2019; Agarwal et al., 2020; Fu et al., 2020; Kumar et al., 2020; Kostrikov et al., 2021), however, are not designed for large discrete action spaces, as they require either maximizing a Q-function or parameterizing a policy over the full discrete action set — operations that become intractable as the space scales exponentially with $\prod_{d=1}^{A} m_d$, where $A$ is the number of sub-action dimensions and $m_d$ is the number of choices per dimension.

Learning in these complex settings requires solving two related problems: (i) searching over an exponential number of joint actions, and (ii) ensuring that the chosen sub-actions form coherent combinations. Methods designed for these combinatorial spaces have traditionally simplified policy learning by imposing strong structural priors such as assuming conditional independence between sub-actions (Tang et al., 2022; Beeson et al., 2024). However, this sacrifices representational capacity, precluding the model from capturing the interactions required for effective control. Other approaches attempt to learn the action representation and optimize a policy simultaneously (Zhang et al., 2018; Landers et al., 2024; 2025), but this conflation of objectives often makes learning slow and unstable.

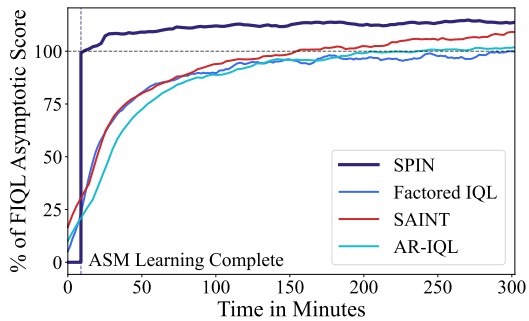

Figure 1: In the `humanoid-stand` task, learning from the `medium-expert` dataset, SPIN reaches the target performance (dashed horizontal line) quickly after pre-training (dashed vertical line), while baselines require over 200 minutes of wall-clock training.

---

*qwp4pk@virginia.edu
[1]Code is available at https://github.com/matthewlanders/SPIN

We introduce **S**tructured **P**olicy **IN**itialization (SPIN), a two-stage framework that decouples representation learning from control. In the first stage, an Action Structure Model (ASM) is trained with self-supervision to learn a representation function that, conditioned on the state $s$, induces a feature space over sub-actions in which structurally coherent joint actions concentrate on a low-dimensional manifold. This action space representation is then frozen during the second stage, where the control problem reduces to learning lightweight policy heads over the action manifold for the downstream RL task. By learning structure first and policy second, SPIN allows the agent to exploit the underlying action geometry instead of searching the raw combinatorial space. This leads to faster training and improved policy performance (Figure 1). Across diverse benchmarks varying in dataset size and quality, action dimensionality, and action cardinality, SPIN **improves average return by up to 39%** over the state of the art and **reduces training time to state-of-the-art performance by up to 12.8×**.

Our contributions are as follows:

- We reframe offline RL in discrete structured action spaces as a representation problem, separating learning action structure from control.
- We propose SPIN, a two-stage framework that pre-trains and freezes an action-space representation to accelerate and improve policy learning.
- We show that SPIN achieves state-of-the-art performance on challenging benchmarks, outperforming existing methods while being significantly faster.
- We analyze the learned representations to demonstrate that capturing action structure is critical for effective policy learning in discrete combinatorial action spaces.

## 2 RELATED WORK

**RL in Large Discrete Action Spaces.** Several RL methods have been developed for combinatorial action spaces in domains such as routing Nazari et al. (2018); Delarue et al. (2020) and resource allocation Chen et al. (2024), but these approaches typically rely on task-specific knowledge. General-purpose methods have also been introduced (Dulac-Arnold et al., 2015; Tavakoli et al., 2018; Farquhar et al., 2020; Van de Wiele et al., 2020; Zhao et al., 2023), but they are typically designed for online learning and are not easily adapted to the constraints of offline datasets. In offline RL, methods often factorize the policy or Q-function (Tang et al., 2022; Beeson et al., 2024). This factorization, however, enforces conditional independence across sub-actions, which restricts representational capacity and fails when sub-actions are strongly dependent. Other methods capture dependencies explicitly — BraVE (Landers et al., 2024) models cross-dimensional interactions but scales poorly with action size, while autoregressive policies (Zhang et al., 2018) impose a fixed ordering that breaks permutation invariance. More recently, SAINT (Landers et al., 2025) introduced a Transformer-based policy to capture sub-action dependencies through self-attention, but learns action structure and control jointly, leading to slow and unstable training. A related line of work learns representations for large, but flat, action spaces. Most relevant is MERLION (Gu et al., 2022), which learns a pseudometric-based action representation for offline RL. However, MERLION's policy execution requires a nearest-neighbor search over the full, enumerated action set at every timestep, making it computationally infeasible for the combinatorial settings we consider. Furthermore, its architecture treats actions as atomic entities and does not model their underlying compositional structure. SPIN, by contrast, is designed for this combinatorial regime, with a structured policy that generates joint actions dimension-wise rather than enumerating the full combinatorial set.

**Self-Supervised Pre-Training in RL.** Self-supervised pre-training in RL has taken several forms including as auxiliary objectives for representation shaping (Jaderberg et al., 2016; Shelhamer et al., 2016), contrastive and predictive encoders (Laskin et al., 2020; Schwarzer et al., 2021; Stooke et al., 2021; Liu & Abbeel, 2021b;a), and world-modeling (Ha & Schmidhuber, 2018). Other work explores masked decision or trajectory modeling (Cai et al., 2023; Liu et al., 2022; Wu et al., 2023; Sun et al., 2023). Large-scale behavioral pre-training has produced generalist policies and vision–language–action models (Brohan et al., 2022; Zitkovich et al., 2023; O'Neill et al., 2024; Kim et al., 2024; Team et al., 2024; Tirinzoni et al., 2025), with methods for rapid post-pre-train adaptation (Sikchi et al., 2025). These approaches are largely state/trajectory-centric and often presume online interaction or multi-task fine-tuning. SPIN, by contrast, pre-trains an Action Structure Model that

captures action composition, providing a structured initialization for policy learning in combinatorial action spaces without any online interaction.

# 3 PRELIMINARIES

An RL problem is formalized as a Markov Decision Process (MDP) $\mathcal{M} = \langle \mathcal{S}, \mathcal{A}, p, r, \gamma, \mu \rangle$, where $\mathcal{S}$ is the state space, $\mathcal{A}$ the action space, $p(s' \mid s, a)$ the transition dynamics, $r(s, a)$ the reward function, $\gamma \in [0, 1]$ the discount factor, and $\mu$ the initial state distribution. A policy $\pi : \mathcal{S} \to \mathbb{P}(\mathcal{A})$ maps states to distributions over actions. The optimal policy maximizes the expected discounted return:

$$\pi^* = \arg\max_\pi \mathbb{E}_\pi \left[ \sum_{t=0}^\infty \gamma^t r(s_t, a_t) \mid s_0 \sim \mu, a_t \sim \pi(\cdot \mid s_t), s_{t+1} \sim p(\cdot \mid s_t, a_t) \right].$$

**Combinatorial action space.** The standard MDP formulation does not define structure in $\mathcal{A}$. In this work, we assume actions are compositional $\mathcal{A} = \mathcal{A}_1 \times \cdots \times \mathcal{A}_N$, with each component $\mathcal{A}_d$ a discrete set of size $m_d$. An action $\mathbf{a} = (a_1, \dots, a_N)$ therefore lies in a space of exponential size $|\mathcal{A}| = \prod_{d=1}^N m_d$, making naive maximization and policy parameterization infeasible.

**Offline reinforcement learning.** Offline reinforcement learning assumes access to a fixed dataset $\mathcal{B} = \{(s_t, a_t, r_t, s_{t+1})\}_{i=1}^Z$ generated by a behavior policy $\pi_\beta$. The learning agent has no additional access to the environment and must train entirely from this static dataset. In offline RL, the maximization in standard temporal difference learning:

$$L(\theta) = \mathbb{E}_{(s,a,r,s') \sim \mathcal{B}} \left[ \left( r + \gamma \max_{a'} Q(s', a'; \theta^-) - Q(s, a; \theta) \right)^2 \right],$$

induces overestimation errors, since actions $a'$ outside the support of $\mathcal{B}$ yield extrapolated values $Q(s', a'; \theta^-)$ without evidence. If such estimates are selected by the $\max$ operator, they are propagated through Bellman updates. Unlike online RL, offline RL cannot correct these errors by interacting with the environment. Reliable offline learning therefore requires constraining policies to the support of $\mathcal{B}$.

# 4 STRUCTURED POLICY INITIALIZATION (SPIN)

**S**tructured **P**olicy **IN**itialization (SPIN) is a two-stage framework for offline RL in structured action spaces that explicitly decouples representation learning from control. In the first stage, an Action Structure Model (ASM) is trained with self-supervision to learn a representation function that, conditioned on the state $s$, induces a feature space over sub-actions in which structurally coherent joint actions concentrate on a low-dimensional manifold. In the second stage, this representation is frozen, and policy learning is reduced to training lightweight heads on the induced action manifold for the downstream RL task.

## 4.1 ACTION STRUCTURE MODELING (ASM)

SPIN's first stage trains an Action Structure Model (ASM) that captures the structure of plausible actions, conditioned on the environment state. Let $f_{\text{ASM}}(s, a; \psi)$ denote a Transformer encoder with parameters $\psi$. The encoder operates on an input sequence $\mathbf{X} \in \mathbb{R}^{(M+N) \times d}$ that concatenates $M$ learned state embeddings $(\mathbf{x}_{s_1}, \dots, \mathbf{x}_{s_M})$ with $N$ sub-action embeddings $(\mathbf{x}_{a_1}, \dots, \mathbf{x}_{a_N})$:

$$\mathbf{X} = (\mathbf{x}_{s_1}, \dots, \mathbf{x}_{s_M}, \mathbf{x}_{a_1}, \dots, \mathbf{x}_{a_N}) \in \mathbb{R}^{(M+N) \times d}.$$

We omit positional encodings across sub-actions to preserve permutation-equivariance over $a_1, \dots, a_N$ (Lee et al., 2019; Landers et al., 2025).

The ASM is trained with a masked conditional modeling objective, analogous to masked language modeling in BERT (Devlin et al., 2019). This objective enables the ASM to capture the manifold of valid actions directly from data, without requiring reward supervision. Because the ASM's role is to learn the manifold before policy optimization begins, this pre-training stage is fundamentally an offline procedure that requires a static, pre-existing dataset. For each $(s, a)$, we sample a subset $\mathcal{M} \subseteq \{1, \dots, N\}$ of sub-action indices to perturb. For every $i \in \mathcal{M}$, $a_i$ is (i) replaced by a mask

---

**Algorithm 1** ASM Pre-Training

---

Initialize parameters $\psi$, heads $\{f_i\}_{i=1}^N$
**for** $t = 1 \dots T_{\text{ASM}}$ **do**
    Sample $(s, a) \sim \mathcal{D}$ with $a = (a_1, \dots, a_N)$
    Sample mask set $\mathcal{M} \subseteq \{1, \dots, N\}$; form $a^{\text{mask}}$ by masking/replacing $a_i$ for $i \in \mathcal{M}$
    Compute attention over sub-actions: $H \leftarrow f_{\text{ASM}}(s, a^{\text{mask}}; \psi)$
    $\mathcal{L}_{\text{ASM}} = \sum_{i \in \mathcal{M}} \text{CrossEntropy}\big(f_i(H_i), a_i\big)$
    Update $\psi, \{f_i\}$ to minimize $\mathcal{L}_{\text{ASM}}$
**end for**
**return** $\psi$

---

token, (ii) replaced by an element drawn uniformly at random from the full sub-action space $\mathcal{A}_i$, or (iii) left unchanged, following an 80/10/10 ratio. The perturbed tuple $a^{\text{mask}}$ is then encoded by $f_{\text{ASM}}$, producing per-slot embeddings $h_{a_i}$ at each slot $i$. Finally, each $h_{a_i}$ is mapped by a slot-specific head $f_i : \mathbb{R}^d \to \mathbb{R}^{|\mathcal{A}_i|}$ to logits over $\mathcal{A}_i$, and cross-entropy loss is computed only on the masked sub-actions:

$$\mathcal{L}_{\text{ASM}} = \mathbb{E}_{(s,a)\sim\mathcal{D}}\Big[ \, \mathbb{E}_{\mathcal{M}} \sum_{i \in \mathcal{M}} \ell\big(f_i(h_{a_i}), a_i\big)\Big].$$

The ASM pre-training procedure is summarized in Algorithm 1. We empirically validate this objective in Appendix C, where we show it outperforms strong generative and discriminative alternatives.

## 4.2 Policy Learning with a Frozen Representation

In the second stage, SPIN performs policy learning on the frozen representation provided by the ASM. The policy network $\pi_\theta$ updates only lightweight components such as the query vectors and output heads, while the ASM remains fixed. This separation preserves the learned action structure and keeps policy optimization tractable.

**Policy Architecture and Training**  SPIN implements the policy $\pi_\theta$ using the SAINT architecture (Landers et al., 2025), which models dependencies among sub-actions while preserving tractability. Specifically, $M$ state embeddings and $N$ learnable action queries are passed through the frozen, permutation-equivariant Transformer from the ASM stage, producing contextualized embeddings $\mathbf{z}_1, \dots, \mathbf{z}_N$. Each embedding $\mathbf{z}_i$ encodes the state, the corresponding action query, and its relations to other sub-actions through shared attention. These embeddings are then passed to sub-action-specific MLP heads $f_i$, which output logits $\ell_i$ over the corresponding sub-actions. The logits parameterize categorical distributions $\pi_\theta(\cdot \mid s, \mathbf{z}_i) = \text{softmax}(\ell_i)$, from which $a_i$ is sampled. The joint policy factorizes over these contextualized distributions:

$$\pi_\theta(a \mid s) = \prod_{i=1}^N \pi_\theta(a_i \mid s, \mathbf{z}_i) \, .$$

As in prior factored approaches (Tang et al., 2022; Beeson et al., 2024), this factorization preserves tractability — rather than optimizing over an exponentially large joint action space, the policy produces $N$ categorical distributions. Unlike purely factored methods that assume conditional independence across sub-actions, SPIN retains cross-dimensional dependencies through shared self-attention, learned during ASM pre-training, to produce contextualized embeddings $\mathbf{z}_i$. The full policy learning procedure is summarized in Algorithm 2.

**Compatibility with Offline RL Methods.**  This design makes SPIN **broadly compatible with actor–critic algorithms** for which the actor update is expressed as weighted log-likelihood maximization over dataset actions:

$$\max_\theta \mathbb{E}_{(s,a)\sim\mathcal{D}} \left[ w_\Phi(s, a) \, \log \pi_\theta(a \mid s) \right] \, ,$$

where $w_\Phi(s, a) \geq 0$ encodes algorithm-specific weights (e.g., advantages or value estimates). This class includes methods such as IQL (Kostrikov et al., 2021) and AWAC (Nair et al., 2020). SPIN also supports selection-based updates, as in BCQ (Fujimoto et al., 2019), where the actor is trained on dataset-supported candidate actions.

---

**Algorithm 2** Policy Learning with Frozen Representation

---

Initialize policy params $\theta$ (queries $\mathbf{Q}$, heads), aux params $\Phi$ (e.g., critic), optimizers
**for** $t = 1 \ldots T_{\mathrm{RL}}$ **do**
    Sample $(s, a, r, s') \sim \mathcal{D}$
    Get contextualized embeddings: $\mathbf{z}_{1:N} \leftarrow f_{\mathrm{ASM}}(s, \mathbf{Q}; \psi_{\mathrm{frozen}})$            ▷ Policy forward pass
    Compute policy loss: $\mathcal{L}_\theta = - w_\Phi(s, a) \sum_{i=1}^{N} \log \pi_\theta(a_i \mid s; \mathbf{z}_i)$
    Compute auxiliary loss $\mathcal{L}_\Phi$ with chosen offline RL objective
    Update both $\theta$ and $\Phi$ with their respective gradients            ▷ Only queries/heads train
**end for**
**return** $\theta$

---

Objectives requiring global operations over the joint action space are intractable unless $Q_\Phi$ or $\pi_\theta$ are factorized across action dimensions, which enforces conditional independence and discards cross-dimensional structure. This conflicts with SPIN's objective of modeling action coherence, so value-regularization methods such as CQL (Kumar et al., 2020) fall outside this compatibility class, mirroring the boundary defined by SAINT.

## 5 EXPERIMENTAL EVALUATION

We evaluate SPIN on a discretized variant of the DeepMind Control Suite (Tassa et al., 2018), introduced by Beeson et al. (2024). The combinatorial action spaces grow exponentially with the number of joints and the sub-action cardinalities. Action spaces vary along two axes: (i) the action dimensions, ranging from six in `cheetah` to 38 in `dog-trot`, and (ii) the sub-action cardinalities, ranging from three to thirty bins per dimension. Together, these variations yield joint action spaces spanning several orders of magnitude, from hundreds to $30^{38} \approx 1.35 \times 10^{56}$ possible actions.

Datasets are constructed at four quality levels following standard offline RL protocols. The `medium` sets are generated by partially trained policies, and the `expert` sets by fully trained policies. The `medium-expert` sets combine transitions from both, while the `random-medium-expert` sets additionally include trajectories with random actions, yielding highly heterogeneous distributions. Dataset sizes range from $2 \times 10^5$ to $2 \times 10^6$ transitions.

We evaluate SPIN against three baselines representing the primary approaches to offline learning in structured action spaces. SAINT (Landers et al., 2025) is a Transformer-based policy architecture that models sub-action interactions through self-attention, jointly learning both the action structure and control policy under a single RL objective. In our evaluation, SAINT serves as the most direct joint-learning comparison to SPIN. We instantiate both methods within the same policy class to ensure that differences in performance reflect the learning paradigm rather than architectural variations. Under this controlled setup, SAINT differs from SPIN precisely in the dimension our work investigates: the separation of representation learning from control. A factored policy (Tang et al., 2022; Beeson et al., 2024) assumes conditional independence across sub-actions, learning separate per-dimension distributions without modeling interactions. An autoregressive policy (Zhang et al., 2018) models dependencies sequentially by factorizing the joint distribution as a chain, conditioning each sub-action on its predecessors. This autoregressive decomposition depends on an arbitrary ordering of dimensions and is not permutation-equivariant.

To isolate the effect of architectural choices, all methods are trained with the IQL (Kostrikov et al., 2021) objective. To assess robustness, we also report results with alternative objectives, including AWAC (Nair et al., 2020) and BCQ (Fujimoto et al., 2019), in Appendix D. To validate SPIN's generalizability beyond locomotion, we evaluate its performance on Maze (Beeson et al., 2024), with results provided in Appendix E. To demonstrate that SPIN's effectiveness is due to its *action-centric* pre-training objective rather than from pre-training alone, we compare its performance to that of a *trajectory-centric* pre-training approach in Appendix F. Across all of these settings, SPIN consistently outperforms the baselines in both performance and efficiency.

All experiments are run using Python 3.9 with PyTorch 2.6 on a single NVIDIA A40 GPU. Reported results are averaged over five random seeds, with $\pm$ values indicating one standard deviation across seeds.

| Task | F-IQL | AR-IQL | SAINT | SPIN |
|------|-------|--------|-------|------|
| **Medium** | | | | |
| cheetah | **293.0** $\pm$ 6.9 | 284.7 $\pm$ 7.2 | **293.5** $\pm$ 6.1 | **293.6** $\pm$ 4.1 |
| finger | 385.4 $\pm$ 6.6 | 383.5 $\pm$ 8.1 | **391.5** $\pm$ 7.7 | **392.7** $\pm$ 8.3 |
| humanoid | **335.4** $\pm$ 6.0 | 327.3 $\pm$ 6.7 | 332.3 $\pm$ 7.2 | **334.8** $\pm$ 7.6 |
| quadruped | 353.4 $\pm$ 82.4 | 343.4 $\pm$ 84.4 | 354.7 $\pm$ 75.1 | **359.7** $\pm$ 78.7 |
| *Average Return* | **341.8** | 334.7 | **343.0** | **345.2** |
| *Time to Target* | **48.2** | 114.3 | 174.6 | **45.5** |
| **Medium-Expert** | | | | |
| cheetah | 612.9 $\pm$ 50.2 | 609.9 $\pm$ 37.7 | 627.4 $\pm$ 37.1 | **651.1** $\pm$ 33.1 |
| finger | 844.5 $\pm$ 11.1 | **857.8** $\pm$ 8.5 | 847.6 $\pm$ 14.6 | **855.2** $\pm$ 9.7 |
| humanoid | 603.0 $\pm$ 49.5 | 567.6 $\pm$ 50.4 | 621.5 $\pm$ 53.5 | **652.5** $\pm$ 31.0 |
| quadruped | 838.2 $\pm$ 45.9 | 833.4 $\pm$ 46.4 | 836.5 $\pm$ 35.9 | **854.1** $\pm$ 37.7 |
| *Average Return* | 724.7 | 717.2 | 733.3 | **753.2** |
| *Time to Target* | 257.3 | 285.8 | 308.4 | **62.0** |
| **Random-Medium-Expert** | | | | |
| cheetah | 289.6 $\pm$ 17.0 | 276.5 $\pm$ 13.2 | 302.4 $\pm$ 33.2 | **332.4** $\pm$ 43.1 |
| finger | 693.4 $\pm$ 17.5 | 762.7 $\pm$ 20.1 | 747.2 $\pm$ 17.1 | **773.2** $\pm$ 14.9 |
| humanoid | 230.1 $\pm$ 21.9 | 192.8 $\pm$ 24.1 | 237.3 $\pm$ 29.4 | **330.2** $\pm$ 28.6 |
| quadruped | 340.0 $\pm$ 52.6 | 350.2 $\pm$ 84.5 | 468.8 $\pm$ 55.5 | **561.0** $\pm$ 70.6 |
| *Average Return* | 388.3 | 395.6 | 438.9 | **499.2** |
| *Time to Target* | 85.1 | 95.8 | 100.2 | **38.4** |
| **Expert** | | | | |
| cheetah | 665.9 $\pm$ 24.2 | 665.7 $\pm$ 20.1 | 664.5 $\pm$ 25.8 | **672.7** $\pm$ 21.5 |
| finger | **874.0** $\pm$ 4.6 | **873.3** $\pm$ 6.7 | 870.2 $\pm$ 9.2 | 868.5 $\pm$ 6.4 |
| humanoid | **729.0** $\pm$ 21.2 | 717.2 $\pm$ 25.0 | 726.1 $\pm$ 25.3 | **734.7** $\pm$ 19.5 |
| quadruped | **843.5** $\pm$ 20.2 | 824.9 $\pm$ 35.8 | 831.6 $\pm$ 43.9 | **839.0** $\pm$ 33.9 |
| *Average Return* | **778.1** | 770.3 | **773.1** | **778.7** |
| *Time to Target* | 167.5 | 288.7 | 261.8 | **77.4** |
| *Overall Average Return* | 558.2 | 554.5 | 572.1 | **594.1** |
| *Overall Time to Target* | 558.1 | 784.6 | 845.0 | **223.3** |

Table 1: Asymptotic performance and training efficiency on DM Control tasks. *Time to Target* denotes the wall-clock minutes required to reach 95% of F-IQL's asymptotic performance, with SPIN times including ASM pre-training. SPIN attains the best overall returns and the highest computational efficiency.

## 5.1 ASYMPTOTIC PERFORMANCE AND TRAINING EFFICIENCY

Table 1 reports final performance and training efficiency across environments and dataset qualities (full learning curves are provided in Appendix A). SPIN achieves consistently higher returns than all baselines and reaches the target performance in less wall-clock time than all baselines.

SPIN achieves the highest overall average return of 594.1, exceeding the next-best baseline, SAINT, at 572.1. Improvements are systematic across the benchmark suite rather than concentrated in individual environments. The advantage is most pronounced in the heterogeneous `medium-expert` and `random-medium-expert` datasets, which represent the most realistic and challenging benchmark settings. On the `random-medium-expert` datasets, SPIN achieves an average return of 499.2, an improvement of more than 13% over the next-best method, SAINT (438.9).

We also measure the wall-clock time, reported as the number of minutes, required for each method to reach 95% of F-IQL's asymptotic performance. F-IQL is a widely adopted state-of-the-art baseline in structured action spaces (Tang et al., 2022; Beeson et al., 2024; Landers et al., 2024), offering both tractability and stable convergence across environments. Using F-IQL as the target enables fair comparison across methods that converge to different return levels, avoiding misleading advantages from terminating early at suboptimal performance. We adopt the 95% threshold rather than 100% because some methods never reach F-IQL's asymptotic performance. Handling these cases directly — either by excluding runs or by reporting full runtimes — would bias the results, whereas the 95%

| Bins | F-IQL | AR-IQL | SAINT | SPIN |
|---|---|---|---|---|
| 3 Bins | $472.3 \pm 43.5$ | $526.5 \pm 57.8$ | $635.1 \pm 39.6$ | $\mathbf{647.0} \pm 19.6$ |
| 10 Bins | $483.8 \pm 33.0$ | $457.4 \pm 53.2$ | $529.1 \pm 58.2$ | $\mathbf{629.5} \pm 52.5$ |
| 30 Bins | $485.0 \pm 54.0$ | $557.4 \pm 66.0$ | $562.5 \pm 88.7$ | $\mathbf{703.9} \pm 25.6$ |
| *Average Return* | 480.4 | 513.8 | 575.6 | **660.1** |
| *Time to Target* | 545.8 | 692.6 | 291.6 | **237.0** |

Table 2: Performance and efficiency on the `dog-trot` task as action cardinality increases. SPIN sustains strong returns and training efficiency as the action space grows, while baselines stagnate or deteriorate.

criterion offers a consistent and comparable metric. Full per-environment time-to-target results are reported in Appendix B. In total, SPIN reaches the target performance in 223.3 minutes, approximately $2.5\times$ faster than F-IQL itself and $3.8\times$ faster than SAINT. The acceleration is especially pronounced in the `medium-expert` datasets, where SPIN requires only 62 minutes of training time compared to more than 250 minutes for all other methods. All runtimes for SPIN include the full duration of the ASM pre-training stage.

These findings demonstrate that explicitly modeling action structure in a dedicated pre-training phase allows the representation to capture the manifold of coherent actions. Freezing this representation during policy learning preserves that structure, enabling lightweight heads to adapt efficiently to the downstream task. Compared to Factored and Autoregressive approaches, which either discard or impose rigid structure on cross-dimensional dependencies, SPIN retains flexibility without sacrificing tractability. Unlike SAINT, which attempts to learn action structure and control jointly, SPIN's decoupled design achieves both higher asymptotic performance and faster convergence.

## 5.2 ROBUSTNESS TO ACTION CARDINALITY

The results in Section 5.1 report evaluations with a fixed action cardinality of three bins per dimension. To test robustness under more severe combinatorial growth, we increased the granularity of discretization in the `dog-trot` environment, which has 38 sub-action dimensions. Varying the cardinality from 3 to 30 bins produces action spaces ranging from $3^{38} \approx 1.35 \times 10^{18}$ to more than $30^{38} \approx 1.35 \times 10^{56}$ possible actions. Experiments were conducted on the `medium-expert` dataset, which provides a realistic and challenging setting.

Results are summarized in Table 2. SPIN achieves the highest average return at every cardinality, and the gap relative to baselines increases with the size of the action space. At three bins, SPIN slightly outperforms the strongest baseline, SAINT. At thirty bins, SPIN reaches an average return of 703.9 compared to 562.5 for SAINT, an improvement of more than 25%. AR-IQL shows unstable performance, dropping from 526.5 at three bins to 457.4 at ten bins, while F-IQL shows no benefit from increased granularity, plateauing around 480.

Training efficiency follows the same trend. SPIN consistently requires less wall-clock time to reach the target performance, even in the largest action spaces (full runtime results are provided in Appendix B). These results demonstrate that separating structure learning from control is increasingly beneficial as combinatorial complexity grows, as the agent can act over a learned low-dimensional manifold while end-to-end methods remain tied to the scale of the raw joint space.

## 6 MECHANISMS UNDERLYING SPIN'S EFFECTIVENESS

The experiments in Section 5 show that SPIN outperforms existing methods both in learning speed and final performance. We now examine the mechanisms underlying these gains.

### 6.1 EFFECT OF REPRESENTATION QUALITY ON POLICY PERFORMANCE

To assess the contribution of ASM pre-training, we trained the ASM representation on the `medium-expert` datasets for 10–100 epochs. Each representation function was then frozen and used to initialize a new policy, which was subsequently trained to convergence on the control task. Figure 2 shows that downstream return generally improves with more ASM pre-training, with the

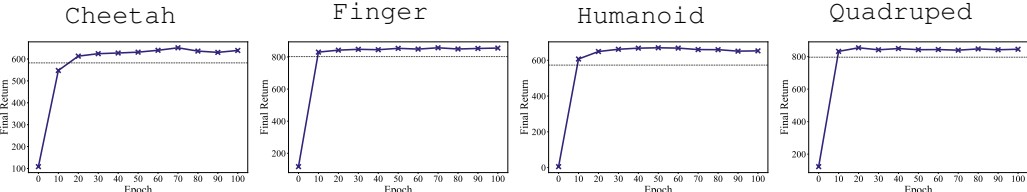

Figure 2: Final policy return as a function of ASM pre-training duration. Policies are initialized from frozen ASMs trained for different numbers of epochs. While pre-training is critical, just 20 epochs yield a representation that enables policies to surpass the fully trained F-IQL baseline in all tasks (dashed horizontal line).

steepest gains in the first 20 epochs. After 20 epochs, policies surpass the fully converged F-IQL reference on all tasks. Policies initialized from an untrained ASM (Epoch 0) perform poorly. These results indicate that final policy performance is largely determined by the quality of the pre-trained action representation; once a coherent representation is learned, control optimization becomes substantially easier.

## 6.2 QUANTIFYING REPRESENTATION QUALITY

The large gap between randomly initialized (epoch 0) and pre-trained agents in Figure 2 may be due to pre-training providing only a convenient initialization, without encoding structure, or due to pre-training learning a representation that enables downstream performance. We evaluate this directly by testing whether the ASM representation captures joint action dependencies using a linear probe, a standard diagnostic for self-supervised representations (Chen et al., 2020; He et al., 2020).

In this experiment, the ASM representation is frozen — either pre-trained for 100 epochs or randomly initialized — and a lightweight linear classifier is trained on its embeddings to predict dataset actions from the state. New action queries and linear heads are learned for this probe. The analysis is conducted in the `dog-trot` environment, which has 38 sub-action dimensions discretized into 30 bins, yielding the largest and most challenging combinatorial action space in the DM Control suite.

The random ASM (0 epochs) achieves 76.6% per-slot accuracy, but its exact-match accuracy on the full 38-dimensional action tuple is only 0.10% — well above the uniform-chance baseline of $30^{-38} \approx 7 \times 10^{-57}$, but far below the level required for coherent control. These features capture individual action dimensions but fail to encode cross-joint dependencies. The pre-trained ASM, by contrast, attains 90.0% per-slot accuracy and 4.52% exact-match accuracy, a $45\times$ improvement over the random ASM. Crucially, the observed tuple accuracy more than doubles the value expected under independence ($0.90^{38} \approx 1.83\%$), showing that pre-training produces a representation with substantial cross-joint coordination.

## 6.3 ISOLATING THE CONTRIBUTION OF THE LEARNED REPRESENTATION

The analyses in Sections 6.1 and 6.2 indicate that final policy performance is largely determined by the quality of the pre-trained action representation. This suggests a sufficiently expressive representation should reduce the architectural demands on the downstream policy, allowing strong performance to be achieved even with a simple policy head. To test this hypothesis directly, we introduce SPIN-Distill, a variant that replaces the Transformer-based policy in Stage 2 (Section 4.2) with a lightweight, attention-free MLP. Specifically, the knowledge from the frozen ASM $f_{\mathrm{ASM}}(s, \mathbf{Q}; \psi_{\mathrm{frozen}})$ is distilled into a student network, $g_\phi(s, \mathbf{Q})$ using a mean-squared error objective to reproduce the contextualized embeddings of the teacher:

$$\mathcal{L}_{\mathrm{Distill}} = \mathbb{E}_{s \sim \mathcal{D}} \left[ \left\| g_\phi(s, \mathbf{Q}) - f_{\mathrm{ASM}}(s, \mathbf{Q}; \psi_{\mathrm{frozen}}) \right\|^2 \right].$$

After training, the student network is frozen and functions as a lightweight, attention-free feature extractor for the downstream policy. Table 3 reports the results of this experiment.

SPIN-Distill comes within a small margin of the full SPIN model's asymptotic performance and substantially outperforms all other baselines, while being nearly $8\times$ faster than SAINT. These results provide strong evidence that SPIN's performance gains are attributable to the quality of the pre-trained representation itself, and not the policy network's specific architecture.

| Task | F-IQL | AR-IQL | SAINT | SPIN | SPIN-Distill |
|------|-------|--------|-------|------|--------------|
| cheetah | $612.9 \pm 50.2$ | $609.9 \pm 37.7$ | $627.4 \pm 37.1$ | $\mathbf{651.1 \pm 33.1}$ | $\mathbf{645.3 \pm 40.8}$ |
| finger | $844.5 \pm 11.1$ | $857.8 \pm 8.5$ | $847.6 \pm 14.6$ | $\mathbf{855.2 \pm 9.7}$ | $\mathbf{852.1 \pm 9.3}$ |
| humanoid | $603.0 \pm 49.5$ | $567.6 \pm 50.4$ | $621.5 \pm 53.5$ | $\mathbf{652.5 \pm 31.0}$ | $\mathbf{650.9 \pm 28.8}$ |
| quadruped | $838.2 \pm 45.9$ | $833.4 \pm 46.4$ | $836.5 \pm 35.9$ | $\mathbf{854.1 \pm 37.7}$ | $\mathbf{846.8 \pm 27.7}$ |
| *Average Return* | 724.6 | 717.2 | 733.3 | **753.2** | **748.8** |
| *Time to Target* | 257.3 | 285.8 | 308.4 | **62.0** | **39.2** |

Table 3: Performance on `medium-expert` datasets. SPIN-Distill achieves asymptotic performance similar to SPIN, indicating that a strong learned representation can compensate for a simpler policy architecture.

### 6.4 EMERGENT RAPID ADAPTATION

Having established the importance of pre-training and representation quality, we next examine learning dynamics. Table 4 reports the percentage of F-IQL's asymptotic performance achieved after 10,000 gradient steps, corresponding to only 1% of the total training budget. Across nearly all environments, SPIN learns policies that reach at least 90% of target performance, whereas baselines improve much more gradually. The effect is most pronounced on heterogeneous datasets. In the `humanoid` task with the `medium-expert` dataset, SPIN reaches 93.4% of target performance, while the next-best method, SAINT, achieves only 9.3%. On the `random-medium-expert` datasets, SPIN exceeds 100% of F-IQL's asymptotic performance in both `cheetah` and `humanoid` during this period.

This rapid learning also clarifies SPIN's wall-clock efficiency (Table 1). The downstream RL stage is computationally dominated by the actor–critic loop, which requires repeated evaluations of the actor, critic, and target networks as well as Bellman backups at every gradient step. The ASM pre-training stage, by contrast, is a stable, single-pass supervised objective applied over masked sub-actions. Its relative cost is therefore minimal: on the `medium-expert` datasets, pre-training constitutes only 5.6% of total wall-clock time on `cheetah`, 1.4% on `finger`, and 1.6% on both `humanoid` and `quadruped`.

Together, these results suggest that the ASM provides a strong structural prior that substantially simplifies downstream learning. End-to-end baselines must jointly discover both action structure and control, leading to slow initial progress, whereas SPIN begins policy learning with a coherent representation, enabling efficient early adaptation and reduced overall training time.

## 7 DISCUSSION AND CONCLUSION

Reinforcement learning in discrete combinatorial action spaces requires searching over an exponential number of composite actions while ensuring that the chosen sub-actions form coherent sets. Some methods simplify policy learning by ignoring action structure (Tang et al., 2022; Beeson et al., 2024), at the cost of discarding critical sub-action dependencies. Other approaches attempt to simultaneously capture structure and solve control (Zhang et al., 2018; Landers et al., 2024; 2025), but are often prohibitively slow and unstable. SPIN, by contrast, separates representation learning from policy learning using a two-stage procedure. In the first stage, an Action Structure Model (ASM) learns a representation function that, conditioned on the state $s$, induces a feature space over sub-actions in which structurally coherent joint actions lie on a low-dimensional manifold. This representation is then frozen and reused in the second stage, where control reduces to training lightweight policy heads on top of the pre-trained ASM.

Across benchmarks varying in dataset size and quality, action dimensionality, and action cardinality, SPIN **improves average return by up to** 39% over the state of the art and **reduces the time to reach strong baseline performance by up to** 12.8×. Gains are most pronounced in the challenging, realistic `medium-expert` and `random-medium-expert` datasets.

A targeted analysis elucidates SPIN's effectiveness. Final performance rises with the quality of the learned representation, confirming that control is bottlenecked by structure discovery. Once this structure is available, policies learn rapidly, reaching most of their eventual return within a small fraction of training. Linear probes further show that the learned representation is 45× more effective

| Task | F-IQL | AR-IQL | SAINT | SPIN |
|---|---|---|---|---|
| **Medium** | | | | |
| cheetah | **94.0**% | 91.0% | **94.6**% | **94.5**% |
| finger | 90.2% | 89.0% | 89.8% | **91.2**% |
| humanoid | 50.9% | 67.3% | 84.9% | **94.2**% |
| quadruped | 90.3% | 82.7% | 92.7% | **94.0**% |
| *Average* | 81.4% | 82.5% | 90.5% | **93.5**% |
| **Medium-Expert** | | | | |
| cheetah | 30.3% | 40.9% | 46.1% | **90.6**% |
| finger | 0.37% | 0.70% | 0.39% | **68.5**% |
| humanoid | 2.2% | 3.1% | 9.3% | **93.4**% |
| quadruped | 34.1% | 38.1% | 39.0% | **78.6**% |
| *Average* | 16.7% | 20.7% | 23.7% | **82.8**% |
| **Random-Medium-Expert** | | | | |
| cheetah | 87.7% | 93.3% | 94.0% | **103.6**% |
| finger | 31.1% | 34.6% | 33.6% | **49.0**% |
| humanoid | 23.3% | 30.4% | 50.5% | **110.7**% |
| quadruped | 43.3% | 48.2% | 45.0% | **80.7**% |
| *Average* | 46.4% | 51.6% | 55.8% | **86.0**% |
| **Expert** | | | | |
| cheetah | 26.5% | 35.8% | 49.1% | **94.3**% |
| finger | 31.6% | 33.1% | 8.4% | **94.9**% |
| humanoid | 2.8% | 2.3% | 14.0% | **95.1**% |
| quadruped | 44.6% | 45.7% | 53.0% | **79.2**% |
| *Average* | 26.4% | 29.2% | 31.1% | **90.9**% |
| *Overall Average* | 42.7% | 46.0% | 50.3% | **88.3**% |

Table 4: Percentage of final F-IQL performance achieved after 10,000 gradient steps (1% of training budget). SPIN quickly learns a high-performing policy, while baselines improve more gradually.

at producing fully coordinated actions than a random baseline, providing a direct, quantitative explanation for the downstream agent's success.

While SPIN demonstrates strong performance, several opportunities for future work remain. Extending SPIN to value-regularization methods such as CQL is a promising direction. One natural next step is to develop hybrid objectives that combine SPIN's representation-first design with mild conservative regularization — for example, penalties restricted to ASM-proposed candidate joint actions or applied at the sub-action level, thereby avoiding intractable global operations over the full combinatorial space. Adapting SPIN to action spaces that exhibit structural assumptions other than permutation equivariance, such as ordered or sequence-based sub-actions is another direction for future work. Finally, as with all offline methods, SPIN's generalization ultimately depends on dataset coverage, and improving robustness under sparse or biased data remains an important open challenge.

SPIN introduces a representation-first view of control in structured action spaces. By first learning the manifold of plausible actions and then reusing a representation function for downstream decision-making, it reduces a complex combinatorial problem to a tractable policy learning task. This decoupling offers a principled framework for reinforcement learning in high–dimensional, structured domains.

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

## A  LEARNING CURVES

Figure 3 shows the full learning curves corresponding to the results in Table 1. Each plot reports mean episode return versus gradient steps for all methods across environments and dataset qualities.

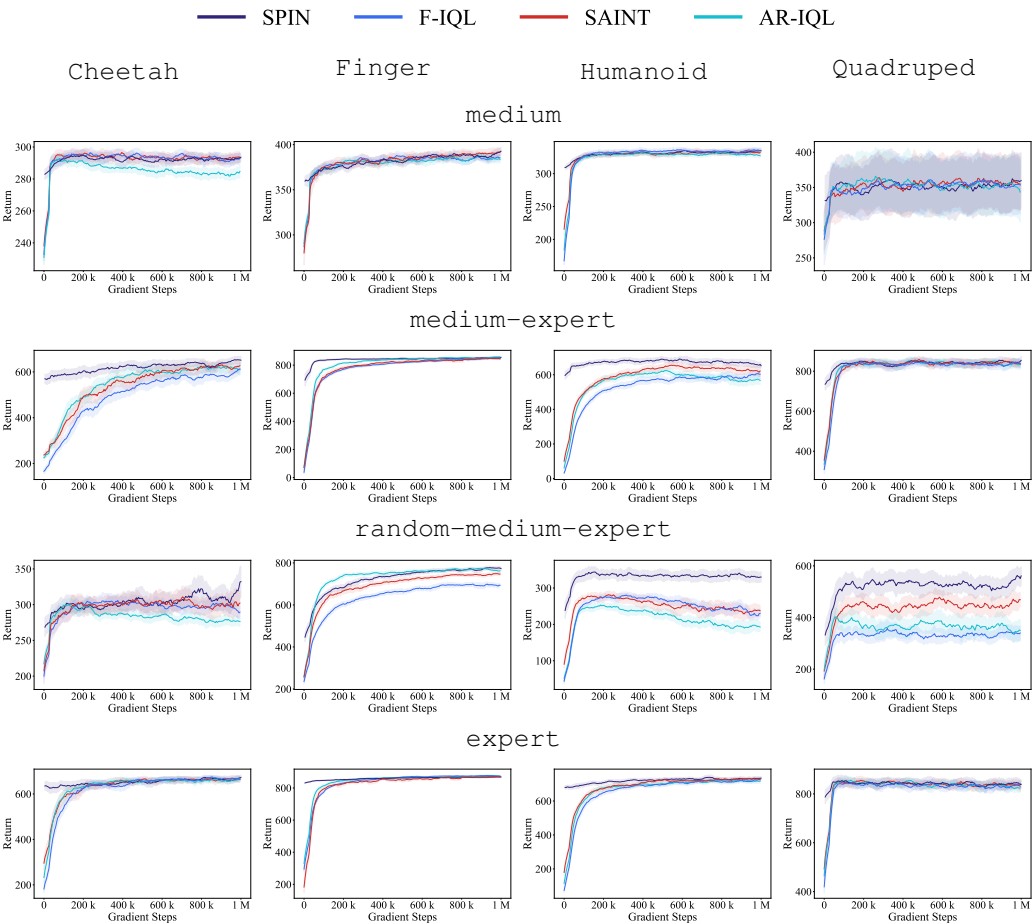

Figure 3: Learning curves on all DM Control tasks and dataset qualities. Curves show mean return over 1M gradient steps, averaged across five seeds with shaded regions denoting ±1 std.

The curves visually confirm the trends reported in the main text — SPIN matches or exceeds the best baseline policy in every environment. SPIN's advantage is most pronounced on the `medium-expert` and `random-medium-expert` datasets, the most challenging and realistic settings.

# B WALL-CLOCK EFFICIENCY

This section reports the full wall-clock training times underlying the efficiency analyses in Sections 5.1 and 5.2. For comparability across methods with different asymptotic returns, we measure the time required to reach 95% of the Factored IQL (F-IQL) baseline's final performance. We adopt the 95% target rather than 100% because not all methods reach F-IQL's asymptote. Alternative treatments — such as excluding these runs or reporting their full runtime — would bias the averages. The 95% criterion provides a consistent point of comparison across methods. All reported SPIN runtimes include the complete ASM pre-training stage.

## B.1 WALL-CLOCK TRAINING TIMES FOR SECTION 5.1

| Task | F-IQL | AR-IQL | SAINT | SPIN |
|---|---|---|---|---|
| **Medium** | | | | |
| cheetah | 7.0 | 13.0 | 12.3 | **2.4** |
| finger | **4.7** | 8.0 | 11.7 | 7.8 |
| humanoid | 25.1 | 59.3 | 43.1 | **23.5** |
| quadruped | **11.4** | 34.0 | 107.5 | **11.8** |
| *Medium Total* | **48.2** | 114.3 | 174.6 | **45.5** |
| **Medium-Expert** | | | | |
| cheetah | 80.0 | 84.1 | 103.3 | **14.9** |
| finger | 28.8 | 20.5 | 56.3 | **14.0** |
| humanoid | 128.2 | 143.2 | 112.2 | **11.2** |
| quadruped | **20.3** | 38.0 | 36.6 | 21.9 |
| *Medium-Expert Total* | 257.3 | 285.8 | 308.4 | **62.0** |
| **Random-Medium-Expert** | | | | |
| cheetah | 11.4 | 13.6 | 17.9 | **3.0** |
| finger | 33.5 | 10.6 | 33.6 | **25.7** |
| humanoid | 23.7 | 55.6 | 33.1 | **5.6** |
| quadruped | 16.5 | 16.0 | 15.6 | **4.1** |
| *Random-Medium-Expert Total* | 85.1 | 95.8 | 100.2 | **38.4** |
| **Expert** | | | | |
| cheetah | 27.6 | 46.3 | 53.3 | **19.6** |
| finger | 16.5 | 16.9 | 38.5 | **4.3** |
| humanoid | 110.0 | 199.6 | 147.3 | **45.6** |
| quadruped | 13.4 | 25.9 | 22.7 | **7.9** |
| *Expert Total* | 167.5 | 288.7 | 261.8 | **77.4** |
| *Total Runtime* | 558.1 | 784.6 | 845.0 | **223.3** |

Table 5: Wall-clock training time (minutes) to reach 95% of F-IQL's asymptotic performance, with SPIN times including ASM pre-training. Italicized rows give totals per dataset quality; the bottom row reports the overall total. This table provides the detailed breakdown for the *Time to Target* results in Table 1.

Table 5 reports the per-environment efficiency results summarized in Table 1. SPIN is consistently the fastest method, with the largest gains on the `medium-expert` and `random-medium-expert` datasets.

## B.2 WALL-CLOCK TRAINING TIMES FOR SECTION 5.2

| Bins | F-IQL | AR-IQL | SAINT | SPIN |
|---|---|---|---|---|
| 3 Bins | 388.3 | 319.3 | 121.7 | **81.2** |
| 10 Bins | 102.1 | 215.2 | 87.9 | **77.8** |
| 30 Bins | **55.4** | 158.1 | 82.0 | 78.0 |
| *Time to Target* | 545.8 | 692.6 | 291.6 | **237.0** |

Table 6: Wall-clock training time (minutes) to target performance in the `dog-trot` environment as action cardinality increases. This table provides the detailed breakdown for the *Time to Target* results in Table 2.

Table 6 reports detailed runtimes for the action cardinality experiments in the `dog-trot` environment, corresponding to Table 2. SPIN is the most efficient method, achieving the lowest total training time across all experiments. Overall, SPIN is the fastest at 3 and 10 bins, while the simpler F-IQL model converges quickest at 30 bins. However, F-IQL's speed comes at the cost of much lower final performance, as shown in Table 2. Crucially, SPIN's runtime remains stable even as the action space expands by many orders of magnitude, demonstrating superior scalability. These results confirm that SPIN offers the best overall balance of strong performance and computational efficiency.

## C  COMPARISON TO ALTERNATIVE PRE-TRAINING OBJECTIVES

To isolate the contribution of the pre-training objective, we compare SPIN's masked modeling to that of strong baselines from the other dominant paradigms in self-supervised learning: a generative objective (*Variational Action Modeling*) and a discriminative objective (*Contrastive Action Modeling*). All experiments are run on the challenging and realistic `medium-expert` datasets.

### C.1  METHODOLOGY

**Masked Action Modeling (MAM).**  This is the reconstructive objective used by SPIN, as described in detail in Section 4.1.

**Variational Action Modeling (VAM).**  As a generative alternative, we adopt a Variational Action Model (VAM) that learns a probabilistic latent representation of plausible actions. The model comprises a Transformer-based encoder $q_\phi(z|s, a)$ and an MLP decoder $p_\theta(a|s, z)$. The encoder reuses the ASM Transformer $f_{\text{ASM}}(s, a; \psi)$ to produce contextualized slot embeddings $(\mathbf{h}_{a_1}, \ldots, \mathbf{h}_{a_N})$, which are mean-pooled to form a global representation $\bar{\mathbf{h}}_a = \frac{1}{N} \sum_{i=1}^N \mathbf{h}_{a_i}$. This representation is mapped to the parameters of a diagonal Gaussian posterior, $\mu_\phi(\bar{\mathbf{h}}_a)$ and $\log \sigma_\phi^2(\bar{\mathbf{h}}_a)$, from which a latent code $\mathbf{z} \in \mathbb{R}^{d_z}$ is sampled using the reparameterization trick (Kingma & Welling, 2013). The decoder $p_\theta$ reconstructs the action by mapping $\mathbf{z}$ (concatenated with a state embedding) to logits for each sub-action dimension via a set of parallel, slot-specific heads $\{f_i : \mathbb{R}^{d_z(+d_s)} \to \mathbb{R}^{|\mathcal{A}_i|}\}_{i=1}^N$. The model is trained by maximizing the Evidence Lower Bound (ELBO):

$$\mathcal{L}_{\text{VAM}} = \mathbb{E}_{(s,a)\sim\mathcal{D}} \left[ \mathbb{E}_{q_\phi(\mathbf{z}|s,a)} \frac{1}{N} \sum_{i=1}^N \log p_\theta(a_i|s, \mathbf{z}) - \beta D_{\text{KL}}\big(q_\phi(\mathbf{z}|s, a) \,\|\, p(\mathbf{z})\big) \right],$$

where $p(\mathbf{z}) = \mathcal{N}(0, \mathbf{I})$ is a standard normal prior and $\beta$ balances reconstruction and regularization.

**Contrastive Action Modeling (CAM).**  We further consider a discriminative objective that learns action structure through contrastive prediction. The Contrastive Action Model (CAM) uses the same Transformer encoder $f_{\text{ASM}}(s, a; \psi)$, trained with a Local-to-Global InfoNCE loss (Oord et al., 2018). The objective encourages each sub-action representation ("local") to be predictable from the joint representation of the remaining sub-actions ("global"). For an unmasked action $(s, a)$, slot embeddings $(\mathbf{h}_{a_1}, \ldots, \mathbf{h}_{a_N})$ are first computed. For each slot $i$, the local token is $\mathbf{h}_{a_i}$ and the global context is the mean of the others, $\mathbf{h}_{\text{ctx},i} = \frac{1}{N-1} \sum_{j \neq i} \mathbf{h}_{a_j}$. These are projected through separate heads $g_{\text{tok}}(\cdot)$ and $g_{\text{ctx}}(\cdot)$ to yield normalized embeddings $\mathbf{q}_i$ and $\mathbf{k}_i$. The model is trained to maximize agreement between matching $(\mathbf{q}_i, \mathbf{k}_i)$ pairs while minimizing agreement with negatives, defined as the context embeddings $\{\mathbf{k}_i^{(b')}\}_{b' \neq b}$ from all other action tuples in the batch. The InfoNCE loss for slot $i$ is defined as:

$$\mathcal{L}_{\text{CAM},i} = -\frac{1}{B} \sum_{b=1}^B \log \frac{\exp(\mathbf{q}_i^{(b)} \cdot \mathbf{k}_i^{(b)}/\tau)}{\sum_{b'=1}^B \exp(\mathbf{q}_i^{(b)} \cdot \mathbf{k}_i^{(b')}/\tau)},$$

where $\tau$ is a temperature hyperparameter (Chen et al., 2020). The overall loss averages across all sub-action positions:

$$\mathcal{L}_{\text{CAM}} = \frac{1}{N} \sum_{i=1}^N \mathcal{L}_{\text{CAM},i}.$$

## C.2 Empirical Comparison

| Task | MAM | VAM | CAM |
|------|-----|-----|-----|
| cheetah | **651.1** $\pm$ 33.1 | 607.5 $\pm$ 54.2 | 607.3 $\pm$ 46.6 |
| finger | **855.2** $\pm$ 9.7 | **858.0** $\pm$ 11.5 | **851.3** $\pm$ 15.9 |
| humanoid | **652.5** $\pm$ 31.0 | 632.3 $\pm$ 51.3 | 638.5 $\pm$ 39.0 |
| quadruped | **854.1** $\pm$ 37.7 | 845.5 $\pm$ 33.4 | 838.5 $\pm$ 44.7 |

Table 7: Mean $\pm$ std policy performance on `medium-expert` tasks under different ASM pre-training objectives.

| Task | MAM | VAM | CAM |
|------|-----|-----|-----|
| cheetah | **14.9** | 46.2 | 64.5 |
| finger | **8.6** | 25.3 | 30.9 |
| humanoid | **39.4** | 135.6 | 243.5 |
| quadruped | **24.4** | 79.7 | 125.4 |

Table 8: Wall-clock pre-training time (minutes) for 100 epochs of pre-training.

Tables 7 and 8 show that MAM yields the strongest or comparable downstream policy performance across all environments and is consistently more efficient to train than both VAM and CAM.

# D ROBUSTNESS TO OFFLINE RL TRAINING OBJECTIVE

To test whether SPIN's benefits extend beyond the IQL objective used in the main experiments, we evaluate it with two additional offline RL methods — Advantage-Weighted Actor-Critic (AWAC) (Nair et al., 2020) and Batch-Constrained Q-Learning (BCQ) (Fujimoto et al., 2019). These experiments are designed to assess whether SPIN can serve as a general framework for accelerating and improving offline RL in structured action spaces. For these experiments, we use the challenging and realistic `medium-expert` datasets.

## D.1 AWAC

| Task | F-AWAC | AR-AWAC | SAINT | SPIN |
|---|---|---|---|---|
| cheetah | $655.6 \pm 33.4$ | $662.1 \pm 24.5$ | $\mathbf{672.5} \pm 19.5$ | $\mathbf{675.6} \pm 17.1$ |
| finger | $1.2 \pm 1.6$ | $2.9 \pm 3.7$ | $681.8 \pm 339.5$ | $\mathbf{863.1} \pm 6.9$ |
| humanoid | $593.4 \pm 93.6$ | $684.0 \pm 34.6$ | $\mathbf{690.5} \pm 31.1$ | $\mathbf{691.7} \pm 32.6$ |
| quadruped | $813.8 \pm 50.3$ | $801.6 \pm 41.3$ | $\mathbf{821.5} \pm 44.0$ | $\mathbf{826.4} \pm 35.7$ |
| **Average** | 516.0 | 537.7 | 716.5 | **764.2** |

Table 9: Mean $\pm$ std performance on DM Control tasks with AWAC variants. SPIN matches or exceeds the performance of the best baseline in every environment.

| Task | F-AWAC | AR-AWAC | SAINT | SPIN |
|---|---|---|---|---|
| cheetah | 68.6 | 125.0 | 122.2 | **97.6** |
| finger | **0.5** | **0.7** | **1.7** | **1.1** |
| humanoid | 84.5 | 82.0 | 72.8 | **3.3** |
| quadruped | 39.6 | 59.0 | 34.3 | **14.6** |
| **Total Runtime** | 193.2 | 266.7 | 231.0 | **116.6** |

Table 10: Wall-clock training time (minutes) to reach 95% of Factored AWAC asymptotic performance, with SPIN times including ASM pre-training.

Tables 9 and 10 report final performance and training efficiency for all architectures under the AWAC objective. SPIN achieves the highest average return (764.2), about 10% higher than the next-best baseline (SAINT). It is also the most efficient, attaining target performance in 116.6 minutes, roughly twice as fast as SAINT.

## D.2 BCQ

| Task | F-BCQ | AR-BCQ | SAINT | SPIN |
|------|-------|--------|-------|------|
| cheetah | $655.2 \pm 33.1$ | $573.9 \pm 111.2$ | $\mathbf{681.3} \pm 21.2$ | $\mathbf{679.2} \pm 23.0$ |
| finger | $690.0 \pm 69.9$ | $753.6 \pm 49.6$ | $834.5 \pm 27.3$ | $\mathbf{850.9} \pm 19.5$ |
| humanoid | $597.2 \pm 45.1$ | $603.2 \pm 49.0$ | $698.4 \pm 43.6$ | $\mathbf{712.4} \pm 31.7$ |
| quadruped | $816.6 \pm 47.3$ | $674.8 \pm 115.8$ | $829.0 \pm 33.7$ | $\mathbf{852.0} \pm 30.0$ |
| **Average** | 689.7 | 651.9 | 760.8 | **773.1** |

Table 11: Mean $\pm$ std performance on DM Control tasks with BCQ variants. SPIN matches or exceeds the performance of the best baseline in every environment.

| Task | F-BCQ | AR-BCQ | SAINT | SPIN |
|------|-------|--------|-------|------|
| cheetah | 29.1 | 33.4 | 30.6 | **18.3** |
| finger | 3.8 | 1.0 | 1.6 | **0.3** |
| humanoid | 80.8 | 81.1 | 31.4 | **4.0** |
| quadruped | 40.8 | 43.7 | 17.0 | **11.6** |
| **Total Runtime** | 154.5 | 159.2 | 80.6 | **34.2** |

Table 12: Wall-clock training time (minutes) to reach 95% of Factored BCQ asymptotic performance, with SPIN times including ASM pre-training.

Tables 11 and 12 show that the trend holds with BCQ as the base algorithm. SPIN attains the highest average performance (773.1). Its efficiency advantage is also pronounced — SPIN reaches target performance in 34.2 minutes, more than twice as fast as SAINT and over four times faster than the factored BCQ variants.

# E  PERFORMANCE ON MAZE

To assess generalization beyond the DM Control locomotion suite, we evaluated all methods on a configurable Maze navigation task introduced in Beeson et al. (2024). Unlike locomotion, this environment emphasizes goal-reaching rather than complex dynamics. Action dimensionality was varied by changing the number of actuators from 5 to 15.

| Num. Actuators | F-IQL | AR-IQL | SAINT | SPIN |
|---|---|---|---|---|
| 5 | $98.2 \pm 2.1$ | $99.4 \pm 0.0$ | $98.3 \pm 2.2$ | $99.4 \pm 0.1$ |
| 10 | $99.4 \pm 0.0$ | $98.4 \pm 2.2$ | $99.4 \pm 0.1$ | $99.4 \pm 0.0$ |
| 15 | $99.4 \pm 0.0$ | $99.4 \pm 0.0$ | $99.4 \pm 0.0$ | $99.4 \pm 0.0$ |
| **Average** | 99.0 | 99.1 | 99.0 | 99.4 |

Table 13: Performance across actuator configurations on the Maze environment. All methods achieve near-expert performance, indicating the task is solvable by all architectural variants.

| Num. Actuators | F-IQL | AR-IQL | SAINT | SPIN |
|---|---|---|---|---|
| 5 | 4.5 | 2.6 | 4.1 | **1.1** |
| 10 | 3.9 | 3.6 | 6.5 | **3.0** |
| 15 | 8.1 | 11.6 | 12.1 | **7.0** |
| **Total Runtime** | 16.5 | 17.8 | 22.7 | **11.1** |

Table 14: Wall-clock training time (minutes) to reach F-IQL asymptotic performance, with SPIN times including ASM pre-training.

Table 13 shows that all methods, including factored and autoregressive baselines, achieve near-expert performance on the Maze task. This suggests that coordination between sub-actions is less demanding than in locomotion tasks and can be captured even without strong sub-action coordination.

Despite similar asymptotic returns, SPIN retains a consistent efficiency advantage. As shown in Table 14 SPIN reaches the F-IQL performance target fastest across all actuator configurations. With a total runtime of 11.1 minutes across experiments, SPIN is about $1.5\times$ faster than the next-quickest baseline (F-IQL).

## F   COMPARISON TO TRAJECTORY-CENTRIC PRE-TRAINING

To validate that SPIN's effectiveness comes from its *action-centric* masked modeling rather than pre-training itself, we compare its performance to that of REPREM (Cai et al., 2023). REPREM also follows a pre-train–finetune paradigm, but like most existing RL pre-training methods (see Section 2), its objective is trajectory-centric, predicting future states and rewards from interleaved state–action sequences. Because RePreM's per-sample cost scales with horizon length, training on the $2 \times 10^6$–transition `medium-expert` datasets is prohibitively expensive. We therefore conduct the comparison on the `random-medium-expert` datasets with $2 \times 10^5$ transitions, which remain heterogeneous and challenging.

| Task | Return | | Runtime | |
|---|---|---|---|---|
| | RePreM | SPIN | RePreM | SPIN |
| cheetah | $272.5 \pm 27.1$ | $\mathbf{332.4} \pm 43.1$ | 585.8 | **1.8** |
| finger | $706.9 \pm 40.0$ | $\mathbf{773.2} \pm 14.9$ | 599.3 | **1.0** |
| humanoid | $180.6 \pm 23.9$ | $\mathbf{330.2} \pm 28.6$ | 580.8 | **4.4** |
| quadruped | $328.8 \pm 95.3$ | $\mathbf{561.0} \pm 70.6$ | 585.7 | **2.9** |
| *Average* | 372.2 | **499.2** | 587.9 | **2.5** |

Table 15: Performance and pre-training runtime on the `random-medium-expert` datasets. SPIN achieves higher returns with over $200\times$ faster training.

Results in Table 15 show that SPIN achieves an average return of 499.2, while REPREM reaches only 372.2, demonstrating that in combinatorial action spaces, modeling the action manifold provides a stronger representation for control than trajectory prediction. The efficiency gap is even larger. SPIN completes training in 10.1 minutes, whereas REPREM requires 2351.6 minutes, more than $200\times$ slower. This difference reflects the horizon-independent design of the ASM.

