# OpenReview forum: "Improving and Accelerating Offline RL in Large Discrete Action Spaces with Structured Policy Initialization"
_ICLR.cc/2026/Conference — ICLR 2026 Poster_

### Official Review · Reviewer_UAbS · 2025-11-01

**Soundness:** 3
**Presentation:** 3
**Contribution:** 3
**Rating:** 8
**Confidence:** 3

**Summary:**

Problem:

The paper tackles offline reinforcement learning in large, discrete combinatorial action spaces, settings where the agent must select from exponentially many joint actions (composed of multiple sub-actions) and ensure these selected sub-actions form coherent combinations. This is relevant for domains like healthcare decision support, robotics, recommendation systems, and fleet management, where online exploration is costly, risky, or infeasible.

Approach:
- The authors propose Structured Policy Initialization (SPIN), a two-stage framework that decouples representation learning from control. :
(a) Action Structure Model (ASM) is trained to learn an action representation function,
(b) ASM is frozen and lightweight policy heads are trained for downstream RL control on this learned action representation.

**Strengths:**

- SPIN offers a principled and empirically validated way to accelerate and improve offline RL in large discrete combinatorial action spaces, primarily by decoupling structure learning and control, thus making learning tractable and robust as complexity grows. The separation of structure and control is motivated and clearly shown to overcome the slowness/instability of joint learning

- SPIN works with multiple offline RL algorithms (IQL, AWAC, BCQ). Overall, SPIN is a promising and elegant approach that reframes discrete combinatorial control as a representation problem.

**Weaknesses:**

- The current framework requires architectural compatibility between ASM and policy modules for effective weight transfer. This can limit its integration with arbitrary RL architectures and restrict broader applicability.

**Questions:**

- The paper notes that SPIN is compatible with IQL and AWAC but not CQL. Could you elaborate on stability issues that arise with value-regularization methods and whether hybrid objectives could reconcile them?

- Why was masked conditional modeling chosen over contrastive or next-sub-action prediction? Did you test alternative pretext tasks, and if so, how did they compare?

- Have you evaluated SPIN on higher-arity combinatorial domains (e.g., VRP or job-shop scheduling) where the sub-action semantics differ? Would the same ASM formulation apply without state–action token alignment?

---

> ### Author Response · Authors · 2025-11-17
>
> Thank you for the thoughtful and encouraging review. We are glad that the motivation, design, and empirical strengths of SPIN came through clearly, and we appreciate your careful reading of the paper. Your questions were especially helpful in highlighting areas where additional clarification and experimentation would strengthen the contribution. We respond to each point below and provide results from **two new experiments** that resolve the questions you raised.
>
> ### W1: Architectural compatibility between ASM and policy
>
> The requirement for compatibility between the ASM and policy arises entirely from our decision to use the SAINT policy architecture. This was a deliberate choice to ensure a controlled comparison. By using the same underlying architecture for both SPIN and the strongest joint-learning baseline (SAINT), we isolate the benefit of our core contribution: the **decoupled, two-stage paradigm**, rather than architectural differences. This reuse, however, is not fundamental to SPIN's framework.
>
> To demonstrate this explicitly, we conducted a **new experiment** introducing SPIN-Distill, which removes the need for any architectural compatibility with SAINT. Motivated by our analysis in Section 6 showing that SPIN's policy performance is determined primarily by the quality of the representation learned during ASM pre-training, we test whether a simplified (i.e., non-SAINT) policy can perform well once the ASM has encoded the action structure.
>
> In SPIN-Distill, we distill the knowledge from the pre-trained ASM into a lightweight, attention-free MLP student network. This MLP serves as a frozen feature extractor for the downstream policy, fully replacing SAINT's Transformer-based policy architecture. We have added a new subsection — Section 6.3, "Isolating the Contribution of the Learned Representation" (line 415) — detailing this experiment.
>
> For the reviewer's convenience, the key results are included below:
>
> | Task              | F-IQL            | AR-IQL           | SAINT            | SPIN                    | SPIN-Distill            |
> | ----------------- | ---------------- | ---------------- | ---------------- | ----------------------- | ----------------------- |
> | cheetah           | $612.9\pm50.2$   | $609.9\pm37.7$   | $627.4\pm37.1$   | $\mathbf{651.1}\pm33.1$ | $\mathbf{645.3}\pm40.8$ |
> | finger            | $844.5\pm11.1$   | $857.8\pm8.5$    | $847.6\pm14.6$   | $\mathbf{855.2}\pm9.7$  | $\mathbf{852.1}\pm9.3$  |
> | humanoid          | $603.0\pm49.5$   | $567.6\pm50.4$   | $621.5\pm53.5$   | $\mathbf{652.5}\pm31.0$ | $\mathbf{650.9}\pm28.8$ |
> | quadruped         | $838.2\pm45.9$   | $833.4\pm46.4$   | $836.5\pm35.9$   | $\mathbf{854.1}\pm37.7$ | $\mathbf{846.8}\pm27.7$ |
> | *Average Return*  | $724.6$          | $717.2$          | $733.3$          | $\mathbf{753.2}$        | $\mathbf{748.8}$        |
> | *Time to Target*  | $257.3$          | $285.8$          | $308.4$          | $62.0$                  | $\mathbf{39.2}$         |
>
> SPIN-Distill comes within a small margin of the full SPIN model's asymptotic performance and substantially outperforms all other baselines, while being nearly 8$\times$ faster than SAINT.
>
> This experiment provides conclusive evidence for SPIN's architectural flexibility — its benefits persist even when replacing the SAINT policy with a lightweight, attention-free head.

---

> ### Author Response · Authors · 2025-11-17
>
> ### Q1: Compatibility with CQL and value-regularization methods
>
> We appreciate the question about this important technical detail.
>
> Our current implementation uses actor objectives of the form:
>
> $$
> \max\_\theta \mathbb{E}\_{(s,a)\sim \mathcal D}\big[w\_\Phi(s,a) \log \pi\_\theta(a\mid s)\big],
> $$
>
> where $w_\Phi(s,a)$ is derived from a critic.
>
> These objectives require only **sample-level operations** on the joint action. Value-regularization methods such as CQL, by contrast, introduce terms that require **global operations over the joint action space**, for example expectations or maxima of $Q_\Phi(s,a')$ under $\pi_\theta$:
>
> $$
> \mathbb{E}\_{a'\sim\pi\_\theta(\cdot\mid s)}[Q\_\Phi(s,a')]
> \quad\text{or}\quad
> \max\_{a'} Q\_\Phi(s,a').
> $$
>
> In combinatorial spaces, computing these quantities for coordinated joint actions is intractable unless $Q_\Phi$ is factorized across dimensions. However, this changes the structure of the actor target, decomposing it into independent per-dimension terms and breaking alignment with SPIN's objective of modeling cross-dimensional dependencies. Importantly, SPIN's learned representation is designed to enable accurate sample-level evaluation of $Q_\Phi(s,a)$ and coordinated joint-action selection; it does not factorize $Q_\Phi$ in the way that would be required to make CQL's global regularization term tractable.
>
> For this reason, we currently focus on objectives that do not rely on global action-space operations. We agree that hybrid objectives are a promising direction and have updated the discussion in Section 7 (line 518) to note this explicitly:
>
> > Extending SPIN to value-regularization methods such as CQL is a promising direction. One natural next step is to develop hybrid objectives that combine SPIN's representation-first design with mild conservative regularization — for example, penalties restricted to ASM-proposed candidate joint actions or applied at the sub-action level, thereby avoiding intractable global operations over the full combinatorial space.

---

> ### Author Response · Authors · 2025-11-17
>
> ### Q2: Justification for pre-training objective
>
> We selected the Masked Action Modeling (MAM) objective (Section 4.1) because masking is a well-established and powerful technique for learning contextual representations, as demonstrated by its success in models like BERT; it is known to be highly stable to train; and it is computationally efficient, as the loss is computed only on a subset of the sub-actions.
>
> However, we agree with the reviewer that alternative self-supervised objectives are also possible. To provide a rigorous empirical justification for our use of masked reconstruction, we conducted a **new experiment** based on your feedback. Specifically, we compare our Masked Action Model (MAM) objective (Section 4.1) to strong alternatives from other major self-supervised paradigms: a Variational Action Model (VAM) and a Contrastive Action Model (CAM).
>
> We provide a full description of the alternative pre-training objectives and the corresponding experimental results in a new Appendix section (Section C). For convenience, we summarize the key findings on the challenging `medium-expert` datasets below:
>
> | Task      | MAM (ours)                  | VAM                     | CAM                     |
> |-----------|-----------------------------|-------------------------|-------------------------|
> | cheetah   | $\mathbf{651.1}\pm33.1$     | $607.5\pm54.2$          | $607.3\pm46.6$          |
> | finger    | $\mathbf{855.2}\pm9.7$      | $\mathbf{858.0}\pm11.5$ | $\mathbf{851.3}\pm15.9$ |
> | humanoid  | $\mathbf{652.5}\pm31.0$     | $632.3\pm51.3$          | $638.5\pm39.0$          |
> | quadruped | $\mathbf{854.1}\pm37.7$     | $845.5\pm33.4$          | $838.5\pm44.7$          |
>
> **Table 7:**  Mean $\pm$ std policy performance on `medium-expert` tasks under different ASM pre-training objectives.
>
> | Task      | MAM (ours)        | VAM    | CAM    |
> |-----------|-------------------|--------|--------|
> | cheetah   | $\mathbf{14.9}$   | $46.2$ | $64.5$ |
> | finger    | $\mathbf{8.6}$    | $25.3$ | $30.9$ |
> | humanoid  | $\mathbf{39.4}$   | $135.6$| $243.5$|
> | quadruped | $\mathbf{24.4}$   | $79.7$ | $125.4$|
>
> **Table 8:** Wall-clock pre-training time (minutes) for 100 epochs of pre-training.
>
> Across all tasks, our MAM objective achieves the strongest or comparable downstream performance while being substantially more efficient than VAM and CAM. These results provide clear empirical evidence that masked action reconstruction is a critical design choice that matches the structure of combinatorial action spaces.
>
> ### Q3: Applicability to higher-arity domains (VRP, job-shop, etc.)
>
> We appreciate this suggestion and agree that VRP and job-shop scheduling are important combinatorial domains. These settings, however, differ in a key structural way from the action spaces targeted in this paper.
>
> SPIN is explicitly designed for per-step action spaces for which components are permutation-equivariant: the sub-action dimensions are symmetric and interchangeable, and the policy should be invariant to how they are indexed. Our ASM exploits this symmetry by treating sub-actions as an unordered set and modeling their interactions via self-attention.
>
> Many VRP and job-shop formulations, by contrast, impose **inherent ordering or temporal structure** within the action itself (e.g., sequence of customers in a route, job schedules). In these domains, permuting sub-actions corresponds to different decisions. Applying SPIN directly would therefore violate the structural assumptions built into our architecture.
>
> However, the underlying idea of offline structure learning before control is not tied to permutation equivariance. Extending SPIN to settings with ordered or graph-structured action components would primarily require replacing the permutation-equivariant ASM with architectures that respect those structures. We have clarified this in Section 7 (line 523):
>
> > Adapting SPIN to action spaces that exhibit structural assumptions other than permutation equivariance, such as ordered or sequence-based sub-actions is another direction for future work.

---

> ### Author Response · Authors · 2025-11-24
>
> We hope our rebuttal addressed your thoughtful questions. We've added clarifications on architectural compatibility, CQL/value-regularization, alternative pre-training objectives, and applicability to higher-arity domains, supported by additional experiments. As we approach the discussion deadline, we'd be happy to clarify any remaining points.

---

### Official Review · Reviewer_rC8Q · 2025-11-01

**Soundness:** 3
**Presentation:** 3
**Contribution:** 2
**Rating:** 4
**Confidence:** 3

**Summary:**

This paper introduces SPIN, a two-stage approach designed to improve efficiency in discrete combinatorial action spaces. Specifically, it separates representation learning from policy learning. In the first stage, an action structure model learns a representation function that captures the manifold of valid actions. In the second stage, this representation is frozen and reused, with lightweight policy heads built on top of the pre-trained action structure model. The experimental results demonstrate clear benefits in terms of both performance and efficiency.

**Strengths:**

The paper is clearly written and well motivated.

The proposed idea is straightforward, and the algorithm is compatible with actor–critic frameworks, which enhances its applicability across a wide range of settings. The experimental results demonstrate the superiority of the proposed approach compared with the three selected baselines.

**Weaknesses:**

While the focus on offline RL is relevant, it is not sufficiently justified in the paper. In particular, SAINT is originally an online approach, which has been used here in an offline setting for comparison. In my view, it is not entirely fair to claim that SAINT jointly learns the action structure and control, as it was designed for a different purpose. This raises questions about the validity of the comparison.

The evaluation is also somewhat limited. The implementation details for the selected baselines are not described clearly, making the fairness of the comparison uncertain. While it is understandable that the authors aimed to keep architectural choices consistent, comparing directly with the original implementations of the baseline methods would strengthen the credibility of the results.

There are a few relevant works in this area that the authors may wish to consider for experimental comparison, such asOHIO (https://openreview.net/forum?id=dTPz4rEDok), and MERLION (https://proceedings.mlr.press/v162/gu22b/gu22b.pdf). Particularly, the paper claims the decoupling the representation learning from control. However, MERLION also learns reusable action embeddings. The contribution over MERLION remains unclear in the paper.

**Questions:**

1. Please justify the experimental comparison.
2. Please clarify the contributions with respect to the earlier works, especially, MERLION.

---

> ### Author Response · Authors · 2025-11-17
>
> Thank you for your review. You raise several important questions regarding SPIN's relationship to prior work. These points can be fully resolved through clarification, which we provide below. We believe these revisions substantively strengthen the paper and hope this will be reflected in your final assessment.
>
> ### W1: Justify focus on offline RL
>
> The offline setting is fundamental to SPIN's two-stage design. The Action Structure Model (ASM) must be pre-trained on a fixed, pre-existing dataset, because its role is to learn the manifold of valid actions *before* any policy learning occurs. This first representation-learning stage enables SPIN to capture the combinatorial structure of the action space, making the subsequent RL stage far simpler and more efficient. Therefore, SPIN inherently requires offline data; the decoupled design cannot be carried out in an online regime where the data are generated by an evolving policy.
>
> We have added the following clarification to Section 4.1 (line 159):
>
> > Because the ASM's role is to learn the manifold before policy optimization begins, this pre-training stage is fundamentally an offline procedure that requires a static, pre-existing dataset.
>
> ### W2/Q1: Fairness of using SAINT as a baseline
>
> We understand the reviewer's concern regarding the comparison to SAINT and provide several clarifications that, together, strongly justify its inclusion.
>
> First, and most importantly, SAINT provides the clearest and most controlled test of our paper's central hypothesis. SAINT exemplifies the **joint-learning paradigm**, where action-structure learning and control are optimized simultaneously under a single RL objective. SPIN, by contrast, introduces a novel **decoupled paradigm** that separates representation learning from control. To ensure a fair comparison, we evaluate both methods within the same policy class, to ensure that performance differences reflect the learning paradigm rather than confounding architectural factors. This makes SAINT the most direct validation of our claim that decoupling leads to superior efficiency and performance.
>
> Second, we would like to clarify the reviewer's concern that SAINT does not jointly learn the action structure and control. SAINT optimizes a single, end-to-end RL objective that simultaneously models sub-action interactions and learns a control policy. This is exactly what we refer to as "joint learning". Among methods that follow this paradigm, SAINT is the state-of-the-art approach for combinatorial action spaces.
>
> Third, SAINT's architecture is fully compatible with offline RL, even though the original paper reports online results. There is nothing inherently "online" about SAINT's design, and using it in an offline setting requires no modification to the architecture.
>
> Finally, SAINT is empirically a very strong baseline. Across our experiments, it is consistently the second-best performing method (behind SPIN) demonstrating that it provides a meaningful and competitive point of reference. Outperforming a strong joint-training method is strong evidence supporting the benefits of our decoupled approach.
>
> To make these points explicit, we have added clarifying text in Section 5 (line 247) of the revised manuscript:
>
> > SAINT is a Transformer-based policy architecture that models sub-action interactions through self-attention, jointly learning both the action structure and control policy under a single RL objective. Although originally evaluated in an online setting, the architecture itself is fully compatible with offline actor updates without modification. In our evaluation, SAINT serves as the most direct joint-learning comparison to SPIN. We instantiate both methods within the same policy class to ensure that differences in performance reflect the learning paradigm rather than architectural variations. Under this controlled setup, SAINT differs from SPIN precisely in the dimension our work investigates: the separation of representation learning from control.

---

> ### Author Response · Authors · 2025-11-17
>
> ### W3/Q1: Implementation details of baselines
>
> We designed our evaluation to ensure a controlled, fair, and reproducible comparison across all methods.
>
> All baselines were implemented within a single, unified, open-source codebase (linked in the paper). This allows us to hold constant every experimental factor *except* the modeling approach to the combinatorial action space — the primary variable under study. Concretely, all methods are trained with the same offline RL objective (IQL in the main results; AWAC and BCQ in Appendix D), on the same datasets, using identical optimizers, learning-rate schedules, and MLP dimensions.
>
> Although the architectural principles of each baseline necessarily differ (factored, autoregressive, or permutation-equivariant), implementing them in a shared codebase ensures that all performance differences reflect the modeling strategy itself rather than unrelated implementation details.
>
> Using original, heterogeneous implementations from prior work would introduce uncontrolled confounds — different training loops, preprocessing pipelines, hyperparameter choices, and evaluation protocols — which would obscure the effect of the action-space modeling choices we attempt to isolate.
>
> ### W4/Q2: Clarifying contributions relative to prior works
>
> Thank you for pointing to these works. We provide a detailed comparison below.
>
> **Comparison to MERLION:**
>
> While MERLION also learns action embeddings, its setting and assumptions are fundamentally different from ours. The key distinctions are:
>
> First, MERLION assumes a *flat, fully enumerable* discrete action set. Its action-selection rule,
>
> $$
> \arg\min_{a\in\mathcal{A}} \|\hat e - \phi(a;s)\|,
> $$
>
> requires computing a distance to every action embedding, which is only feasible when the full action catalog can be explicitly listed (e.g., a few thousand actions, as in MERLION's experiments). This is impossible in our regime, where the joint action space grows combinatorially and cannot be tractably enumerated.
>
> Second, MERLION treats each action as an atomic entity and therefore cannot represent or exploit the compositional structure of combinatorial action spaces. SPIN, by contrast, explicitly represents a joint action as a collection of sub-actions, with the ASM's self-attention layers learning the dependencies across these sub-actions from the offline dataset. By capturing the underlying combinatorial structure of the action space, rather than embedding each action independently, SPIN scales naturally to combinatorially large joint action spaces.
>
> These conceptual differences are reflected directly in MERLION's empirical scope: all of its experiments operate on flat, enumerable action sets. It has not been evaluated in settings where the action space is combinatorial or where modeling structure across sub-actions is required. Our work focuses precisely on this regime, which is outside MERLION's intended domain.
>
> To address this point clearly in the paper, we have updated the Related Work Section (line 91) to include the following:
>
> > A related line of work learns representations for large, but flat, action spaces. Most relevant is MERLION, which learns a pseudometric-based action representation for offline RL. However, MERLION's policy execution requires a nearest-neighbor search over the full, enumerated action set at every timestep, making it computationally infeasible for the combinatorial settings we consider. Furthermore, its architecture treats actions as atomic entities and does not model their underlying compositional structure. SPIN, by contrast, is designed for this combinatorial regime, with a structured policy that generates joint actions dimension-wise rather than enumerating the full combinatorial set.
>
> **Comparison to OHIO:**
>
> While the work provides valuable insights into policy optimization, it does not address large or combinatorial action spaces, nor does it propose mechanisms for scalable action selection. These capabilities are central to our setting. Because OHIO operates in a fundamentally different problem setting, it is orthogonal to the challenges SPIN is designed to address.

---

> ### Author Response · Authors · 2025-11-24
>
> This is a kind reminder that our rebuttal has been submitted. We've clarified our focus on offline RL, justified the experimental comparisons, and provided detailed comparisons to related work such as MERLION. As the discussion deadline approaches, we'd love to hear back from you to make sure we've fully addressed your concerns.

---

### Official Review · Reviewer_98NV · 2025-11-06

**Soundness:** 2
**Presentation:** 2
**Contribution:** 2
**Rating:** 4
**Confidence:** 3

**Summary:**

The authors proposed an algorithm for RL with combinatorial action spaces. The proposed method has two stages. We learn the action structure during the first stage, and then we learn  a policy in the second stage.

**Strengths:**

By separating the learning of action structure and policy, the proposed algorithm overcomes the computational cost issue that a previous work named SAINT has.

**Weaknesses:**

Determining when to finish pretraining and move on to policy training is crucial. Stopping pretraining too early could lead to poor action structure modeling (Sec 6.1 illustrates the importance of sufficient pretraining), and stopping pretraining too late could lead to the same computational cost issue that is with SAINT. The authors do not provide an approach to choose the stopping time of pretraining.

The proposed method largely reuses the policy architecture in SAINT, and thus the novelty of this work is limited.

**Questions:**

Could the authors provide an approach to choose the stopping time of pretraining?

---

> ### Author Response · Authors · 2025-11-17
>
> Thank you for your feedback and for recognizing the strength of our approach. Your primary questions — (1) determining pre-training duration and (2) the relationship between SPIN and SAINT can both be addressed through clarification and new experimental evidence. In response, we added a **new analysis** that explicitly quantifies the cost of pre-training, and conducted a **new experiment** demonstrating that SPIN's benefits arise from the learned representation itself, independent of SAINT's policy architecture. We hope these clarifications warrant a reconsideration of your score.
>
> ### W1/Q1: Duration of pre-training
>
> This is an important and practical question. We address the two potential failure modes: stopping pre-training too early and stopping too late.
>
> **1. Stopping pre-training too early:**
>
> Section 6.1 and Figure 2 already offer direct guidance on the required pre-training duration: downstream return improves sharply in the first few epochs and then plateaus. In particular, **just 20 epochs** of Action Structure Model (ASM) pre-training is sufficient for the resulting policy to surpass the fully converged performance of the strong factored baseline. This shows that the representation is learned efficiently and that SPIN is not overly sensitive to the precise stopping point.
>
> **2. Stopping pre-training too late:**
>
> The reviewer's concern about pre-training becoming as costly as joint training (as in SAINT) does not arise in practice. Avoiding exactly this inefficiency is actually a primary advantage and central contribution of SPIN. As shown in the revision, the ASM pre-training stage constitutes only a **small fraction** of overall wall-clock time — on average just 2.6\% on the challenging `medium-expert` datasets.
>
> To make the computational advantage of SPIN over joint learning explicit, we added a **new analysis** to Section 6.4 (line 454):
>
> > This rapid learning also clarifies SPIN's wall-clock efficiency (Table 1). The downstream RL stage is computationally dominated by the actor–critic loop, which requires repeated evaluations of the actor, critic, and target networks as well as Bellman backups at every gradient step. The ASM pre-training stage, by contrast, is a stable, single-pass supervised objective applied over masked sub-actions. Its relative cost is therefore minimal: on the `medium-expert` datasets, pre-training constitutes only 5.6\% of total wall-clock time on `cheetah`, 1.4\% on `finger`, and 1.6\% on both `humanoid` and `quadruped`.

---

> ### Author Response · Authors · 2025-11-17
>
> ### W2: Architectural novelty and reuse of SAINT
>
> Our decision to use the SAINT policy architecture was a deliberate choice to ensure a controlled comparison. By using the same underlying architecture for both SPIN and the strongest joint-learning baseline (SAINT), we isolate the benefit of our core contribution: the **decoupled, two-stage paradigm**, rather than architectural differences. This reuse, however, is not fundamental to SPIN's framework.
>
> To demonstrate this explicitly, we conducted a **new experiment** introducing SPIN-Distill, which removes the need for any architectural compatibility with SAINT. Motivated by our analysis in Section 6 showing that SPIN's policy performance is determined primarily by the quality of the representation learned during ASM pre-training, we test whether a simplified (i.e., non-SAINT) policy can perform well once the ASM has encoded the action structure.
>
> In SPIN-Distill, we distill the knowledge from the pre-trained ASM into a lightweight, attention-free MLP student network. This MLP serves as a frozen feature extractor for the downstream policy, fully replacing SAINT's Transformer-based policy architecture. We have added a new subsection — Section 6.3, "Isolating the Contribution of the Learned Representation" (line 415) — detailing this experiment.
>
> For the reviewer's convenience, the key results are included below:
>
>
> | Task              | F-IQL            | AR-IQL           | SAINT            | SPIN                    | SPIN-Distill            |
> | ----------------- | ---------------- | ---------------- | ---------------- | ----------------------- | ----------------------- |
> | cheetah           | $612.9\pm50.2$   | $609.9\pm37.7$   | $627.4\pm37.1$   | $\mathbf{651.1}\pm33.1$ | $\mathbf{645.3}\pm40.8$ |
> | finger            | $844.5\pm11.1$   | $857.8\pm8.5$    | $847.6\pm14.6$   | $\mathbf{855.2}\pm9.7$  | $\mathbf{852.1}\pm9.3$  |
> | humanoid          | $603.0\pm49.5$   | $567.6\pm50.4$   | $621.5\pm53.5$   | $\mathbf{652.5}\pm31.0$ | $\mathbf{650.9}\pm28.8$ |
> | quadruped         | $838.2\pm45.9$   | $833.4\pm46.4$   | $836.5\pm35.9$   | $\mathbf{854.1}\pm37.7$ | $\mathbf{846.8}\pm27.7$ |
> | *Average Return*  | $724.6$          | $717.2$          | $733.3$          | $\mathbf{753.2}$        | $\mathbf{748.8}$        |
> | *Time to Target*  | $257.3$          | $285.8$          | $308.4$          | $62.0$                  | $\mathbf{39.2}$         |
>
> SPIN-Distill comes within a small margin of the full SPIN model's asymptotic performance and substantially outperforms all other baselines, while being nearly 8$\times$ faster than SAINT.
>
> This experiment provides conclusive evidence for two claims that directly address the reviewer’s concern about reliance on SAINT's architecture:
>
> 1. SPIN's core contribution is the learned representation, not the inherited policy architecture.
> 2. SPIN is architecturally flexible — its benefits persist even when replacing the SAINT policy with a lightweight, attention-free head.

---

> ### Author Response · Authors · 2025-11-24
>
> This is a friendly reminder that our rebuttal has been submitted. We've added analysis directly addressing pre-training duration and a new experiment demonstrating architectural flexibility. With the discussion deadline approaching, we'd love to hear back from you to ensure we've addressed your feedback.

---

### Official Review · Reviewer_Mkwm · 2025-11-09

**Soundness:** 3
**Presentation:** 3
**Contribution:** 2
**Rating:** 4
**Confidence:** 3

**Summary:**

The authors propose an RL method to handle large discrete action spaces. The method pretrains a transformer with masked action inputs to reconstruct the action, then do RL on top of the learned transformer representation. They show this method matches other large action RL baselines in just a fraction of their training times. The method is evaluated in a modified version of DM Control where the action space is discretized.

**Strengths:**

- The method is very simple and intuitive. By doing self supervised learning on the large action space, one can learn a more meaningful action representation than the original one. This will lead to large downstream gains.
- The paper is well written and easy to understand.
- The experiment section has interesting analysis results to pin down why SPIN is helpful.

**Weaknesses:**

### Empirical evaluation feels toy and contrived
- these methods are all evaluated in rather artificial RL tasks (hopper, quadruped, etc.), where they take a popular benchmark (DM Control) and then factorize the action space. While useful for fast iteration and initial scientific insight, it is insufficient for convincing me that this method, or even the problem of large discrete action space, is useful. The authors motivated the problem by citing natural problems with large action spaces like recommender systems, robot assembly, etc. Could the authors show results in a more realistic problem setting?


### Method novelty
- In terms of methodological novelty, there's not too much at the high level. When you have noisy or high dimensional data, e.g. noisy sensors, high dim images, representation learning is the first thing we try to improve the signal to noise ratio of our data. So doing this for actions, using a standard masked reconstruction objective, seems very obvious, and not too "novel". On the other hand, this method is "novel" in the sense of applying the masked reconstruction objective to this particular problem where action spaces are large.
- This can be seen as a special case of literature studying masked transformers for decision making problems [1], where the mask of the transformer is just set to the action modality. It would be interesting to compare SPIN against a masked transformer baseline that does masking over both state and action modalities.

[1] Masked Trajectory Models for Prediction, Representation, and Control

[2] PASTA: Pretrained Action-State Transformer Agents

**Questions:**

See weaknesses, I would like to see more realistic experiments. For method novelty, it would be addressed by better framing, and also comparing against a standard representation learning approach like masked reconstruction over the entire input sequence.

---

> ### Author Response · Authors · 2025-11-17
>
> Thank you for your thoughtful review. We appreciate your positive comments on SPIN's simplicity, clarity, and analytical depth. Your main concerns focus on (1) the realism of our experimental evaluation and (2) the degree of methodological novelty. Both concerns arise primarily from framing rather than from limitations of the method. We address these directly in the revision, supported by a **new experiment** that makes the clarifications explicit. These additions directly resolve your concerns and strengthen the paper; we hope this will be reflected in your final assessment.
>
> ### W1/Q1: Choice of benchmark environment
>
> Our evaluation uses a discretized variant of the DeepMind Control Suite, which provides the same underlying physics-based locomotion tasks of the de facto standard benchmark for offline RL (D4RL) [1-5]. This discretized formulation is the established benchmark for studying large, discrete, combinatorial action spaces [6]: it preserves the dynamics, rewards, and task structure of the canonical D4RL environments while introducing the combinatorial explosion that motivates SPIN. Because our paper introduces a novel concept — decoupling action-structure learning from control — we intentionally grounded the empirical study in a **public, well-understood, and widely used benchmark**. This ensures SPIN’s efficacy is evaluated in a setting for which results are straightforward to interpret.
>
> As detailed in Section 5, this benchmark allows us to vary:
>
> * Dataset expertise (`medium`, `expert`, `medium-expert`, `random-medium-expert`),
> * Dataset size (2e5–2e6 transitions),
> * Action dimensionality (6–38 sub-actions), and
> * Sub-action cardinality (3–30 bins per dimension)
>
> These choices yield extremely large joint action spaces — up to $30^{38}$ ($\approx 1.35\times 10^{56}$) possible actions — while maintaining the structure and difficulty of the standard D4RL control tasks. This makes the benchmark both realistic and scientifically appropriate for isolating the challenges associated with large discrete action spaces.
>
> ### W2/Q2: Action-centric vs. trajectory-centric pre-training
>
> We agree that masked modeling is a well-established technique. The contribution of SPIN is not the use of masking per se, but its **action-centric formulation**, which is non-obvious and empirically distinct from standard trajectory-level masking.
>
> SPIN differs from prior works: while existing masked transformers for decision making apply masking to **entire trajectories** of $(s, a, \dots)$ tokens, SPIN instead applies masked modeling to the **internal structure of a single joint action vector** at one timestep, $a = (a_1, \dots, a_N)$. This design directly targets the combinatorial structure of the action space rather than modeling temporal structure.
>
> The baseline requested by the reviewer — masked modeling applied to $(s,a)$ trajectories — was also **already implemented and evaluated via RePreM** [7] (Appendix F). The results of this experiment are conclusive:
>
> * **SPIN achieves a 34% higher average return than RePreM (499.2 vs. 372.2).**
> * **SPIN is over 230× faster to train (10.1 minutes vs. 2351.6 minutes).**
>
> These results show that SPIN’s action-centric formulation is fundamental to its performance and efficiency. We clarified this point in the manuscript by adding the following (line 264):
>
> > To demonstrate that SPIN's effectiveness is due to its *action-centric* pre-training objective rather than from pre-training alone, we compare its performance to that of a *trajectory-centric* pre-training approach in Appendix F.

---

> ### Author Response · Authors · 2025-11-17
>
> ### W3/Q3: Use of masking objective for reconstructing actions
>
> To further justify our use of masked reconstruction as a principled design choice motivated by the structure of combinatorial action spaces, and not an obvious default, we conducted a **new experiment** based on your feedback. Specifically, we compare our Masked Action Model (MAM) objective (Section 4.1) to strong alternatives from other major self-supervised paradigms: a Variational Action Model (VAM) and a Contrastive Action Model (CAM).
>
> We provide a full description of the alternative pre-training objectives and the corresponding experimental results in a new Appendix section (Section C). For convenience, we summarize the key findings on the challenging `medium-expert` datasets below:
>
> | Task      | MAM (ours)                  | VAM                     | CAM                     |
> |-----------|-----------------------------|-------------------------|-------------------------|
> | cheetah   | $\mathbf{651.1}\pm33.1$     | $607.5\pm54.2$          | $607.3\pm46.6$          |
> | finger    | $\mathbf{855.2}\pm9.7$      | $\mathbf{858.0}\pm11.5$ | $\mathbf{851.3}\pm15.9$ |
> | humanoid  | $\mathbf{652.5}\pm31.0$     | $632.3\pm51.3$          | $638.5\pm39.0$          |
> | quadruped | $\mathbf{854.1}\pm37.7$     | $845.5\pm33.4$          | $838.5\pm44.7$          |
>
> **Table 7:**  Mean $\pm$ std policy performance on `medium-expert` tasks under different ASM pre-training objectives.
>
> | Task      | MAM (ours)        | VAM    | CAM    |
> |-----------|-------------------|--------|--------|
> | cheetah   | $\mathbf{14.9}$   | $46.2$ | $64.5$ |
> | finger    | $\mathbf{8.6}$    | $25.3$ | $30.9$ |
> | humanoid  | $\mathbf{39.4}$   | $135.6$| $243.5$|
> | quadruped | $\mathbf{24.4}$   | $79.7$ | $125.4$|
>
> **Table 8:** Wall-clock pre-training time (minutes) for 100 epochs of pre-training.
>
> Across all tasks, our MAM objective achieves the strongest or comparable downstream performance while being substantially more efficient than VAM and CAM. These results provide clear empirical evidence that masked action reconstruction is not an obvious default, but a critical design choice that matches the structure of combinatorial action spaces.
>
> [1] Kostrikov et al., "Offline Reinforcement Learning with Implicit Q-Learning"
>
> [2] Fujimoto et al., "A Minimalist Approach to Offline Reinforcement Learning"
>
> [3] Fujimoto et al., "Off-Policy Deep Reinforcement Learning without Exploration"
>
> [4] Chen et al., "Decision Transformer"
>
> [5] Janner et al., "Offline Reinforcement Learning as One Big Sequence Modeling Problem"
>
> [6] Beeson et al., "An Investigation of Offline Reinforcement Learning in Factorisable Action Spaces"
>
> [7] Cai et al., "RePreM: Representation Pre-training with Masked Model for Reinforcement Learning"

---

> ### Author Response · Authors · 2025-11-24
>
> Just a kind reminder that we've posted our response to your review. In our rebuttal, we've added new experiments on alternative pre-training objectives, clarified our benchmark choice, and highlighted our existing comparison to a trajectory-centric method showing SPIN's superior performance and efficiency. Given that the discussion deadline is approaching, we'd love to hear back from you in case you have any lingering concerns.

---

### Author Response · Authors · 2025-11-17

### General response

We sincerely thank all reviewers for their thoughtful and constructive feedback. We are encouraged that they recognized the merits of our work. Reviewers highlighted the strength of our method, describing it as **"principled and empirically validated"** [UAbS], **"very intuitive"** [Mkwm], and a **"promising and elegant approach"** [UAbS]. They emphasized that the separation of structure learning and control is **"well motivated"** [rC8Q], **effectively overcomes the slowness and instability of joint learning** [UAbS], and that it **addresses the computational cost issues** present in prior methods [98NV]. Reviewers also praised the clarity of the paper, calling it **"well written and easy to understand"** [Mkwm] and **"clearly written and well motivated"** [rC8Q], and found the experimental section to contain **"interesting analysis"** [Mkwm] with results that demonstrate **"clear benefits in both performance and efficiency"** [rC8Q].

Reviewers raised several important points, which we address comprehensively in our point-by-point responses. In this general remark, we summarize the two new experiments, new analysis, and clarifications included in the revised manuscript.

Based on the reviewers' feedback, we conducted **two new sets of experiments** and performed **additional analysis** to:

* **Isolate the contribution of the learned representation and show that SPIN's benefits are not tied to the SAINT architecture** by distilling the ASM into a lightweight, attention-free MLP policy head.
* **Empirically justify our choice of a masked modeling pre-training objective** by comparing it against strong alternatives from both the generative and discriminative paradigms.
* **Quantify the computational cost of SPIN’s two stages**, demonstrating that the ASM pre-training phase introduces only minimal overhead while substantially reducing the duration of the far more expensive RL stage.

These additional experiments and analyses further strengthen our confidence in SPIN's effectiveness.

We address the key themes raised by the reviewers through the following revisions and clarifications in the updated manuscript:

* We clarify that our core novelty lies in the two-stage separation of structure learning and control, which isolates the hardest component of combinatorial decision-making — the discovery of the action manifold — and thereby renders the subsequent policy optimization stage simple and efficient [Mkwm, 98NV, rC8Q].
* We clarify the principled nature of our experimental design, justifying our choice of benchmark and the fairness of our baseline comparisons [Mkwm, rC8Q].
* We provide new results that address concerns about architectural rigidity and the choice of pre-training objective, confirming that SPIN is a robust and flexible framework for simplifying complex control problems [98NV, UAbS, Mkwm].

### Notes to reviewers:

Please note that we use W# to address weaknesses and Q# to address questions in our individual responses. All line numbers refer to the updated version of the manuscript uploaded with the rebuttal.

We sincerely thank the reviewers again for their insightful comments, which have helped us sharpen the presentation and strengthen the contributions of the paper. If any concerns remain, we would be grateful for the opportunity to address them further.

---

### Author Response · Authors · 2025-12-01
**Concise Summary of Rebuttal and Revisions**

Dear Area Chair and Reviewers,

We sincerely thank the Area Chair for their service and the reviewers for their constructive and thoughtful feedback. A concise summary of our rebuttal and revisions is provided below.


## Concerns & Suggestions (Method)

| **Concern**                                                                                          | **Reviewer(s)**      | **Our Action**                                                                                                                                                                                                    |
| ---------------------------------------------------------------------------------------------------- | -------------------- |  -------- |
| **Architectural Dependence:** Is SPIN tied to SAINT?                                                 | 98NV (W2), UAbS (W1) | **New Experiment:** Removed SAINT entirely and substituted a simple MLP head; SPIN achieved SOTA performance, proving that SAINT is unnecessary and that gains come from SPIN's learned representation (Sec. 6.3).|
| **Pre-training Objective:** Why masked modeling instead of alternatives? | UAbS (Q2), Mkwm (W2) | **New Experiment:** Compared masked, variational, and contrastive action modeling, finding masking achieves the strongest performance while being significantly faster (Appendix C).                                |
| **Methodological Distinction:** How does SPIN differ from sequence-level masked modeling?            | Mkwm (W2)            | **Clarification:** Already evaluated the trajectory-masking baseline RePreM, showing SPIN achieves **34% higher return** and is **$230\times$ faster**, demonstrating the necessity of action-centric pre-training (Appendix F). |
| **Stopping Pre-training Early:** When to stop ASM pre-training?                     | 98NV (W1)            | **Analysis:** Identified rapid performance plateau, with **20 epochs** (<9 minutes in all envs) already surpassing fully trained baselines (Sec. 6.1).                                                                       |
| **Stopping Pre-training Late:** Could pre-training become as expensive as joint learning?            | 98NV (W1)            | **New Analysis:** Quantified that pre-training constitutes only $\approx$2.6% of total runtime, confirming it does not approach joint-training cost (Sec. 6.4).                                                           |
| **Compatibility with CQL:** What causes stability issues with CQL?             | UAbS (Q1)            | **Clarification:** Explained that CQL requires global expectations over the combinatorial action space, which are intractable; outlined hybrid conservative objectives as promising future work (Sec. 7).         |

---

## Concerns & Suggestions (Evaluation)

| **Concern**                                                                 | **Reviewer(s)** | **Our Action**                                                                                                                                                                                               |
| --------------------------------------------------------------------------- | --------------- | -------- |
| **Experimental Realism:** Is DM Control too artificial?                     | Mkwm (W1)       | **Clarification:** Explained that discretized DeepMind Control is the *established benchmark* for combinatorial action spaces, preserving D4RL dynamics while inducing exponential action growth and enabling controlled experimental variation (Sec. 5). |
| **Fairness of SAINT Comparison:** Is SAINT an appropriate baseline?         | rC8Q (W1/W2)    | **Clarification:** Established SAINT as the strongest joint-learning baseline and the most direct test of SPIN's decoupled paradigm (Sec. 5).   |
| **Relation to prior work:** How does SPIN differ from MERLION / OHIO?                       | rC8Q (W3)       | **Clarification:** Explained that MERLION requires enumerating all actions (infeasible in our setting) and lacks compositional modeling; OHIO addresses a different problem setting (Sec. 2).|
| **Baseline Implementation Consistency:** Were baselines implemented fairly? | rC8Q (W2)       | **Clarification:** Confirmed that all baselines were reimplemented in a unified codebase with identical training infrastructure to ensure controlled comparison (Sec. 5).                                             |

We have incorporated these new experiments, analyses, and clarifications into the revised manuscript (highlighted in blue). We are confident that the updates address the reviewers' concerns and further demonstrate that SPIN is an effective, scalable, and principled approach to control in large combinatorial action spaces.

---

### Meta-Review · Area_Chair_d4VM · 2026-01-07

**Summary:**

This paper introduces a two-stage offline RL framework for large combinatorial action spaces. The core idea is to pre-train an action structure model with a masked sub-action reconstruction objective to capture the manifold of coherent joint actions, then freeze this representation and train lightweight policy heads for downstream control.

The empirical results on discretized DM Control (and the additional analyses in the revision) collectively indicate that separating structure learning from control learning yields a consistent speedup in convergence and improved returns, particularly on the more heterogeneous dataset regimes.

The revision also includes targeted ablations aimed at isolating the source of gains (representation vs. architecture) and justifying key design choices (masked objective vs. alternatives).

Overall, the paper is well motivated and the experimental/analysis package is stronger after the rebuttal. The remaining discussion is primarily about (1) generalization of the benchmarks and (2) methodological novelty relative to existing masked modeling and pretrain–finetune paradigms.

**Reviewer Concerns:**

Concerns addressed

* architectural dependence (98NV, UAbS) -- The revision added an ablation study (SPIN-Distill) to directly address this concern.
* early stop for ASM pre-training (98NV) -- added new pre-training-duration vs. downstream-return curve showing rapid early plateau and a cost breakdown arguing pre-training is a small fraction of total runtime in the reported regimes
* why masked modeling” (UAbS, Mkwm) -- added comparison against alternative objectives (variational / contrastive action modeling) provides empirical evidence that the chosen objective is competitive in performance and more efficient to train.
“difference from trajectory-centric masked modeling” (Mkwm) -- now with direct comparison to a trajectory-centric pretrain–finetune baseline
* related work positioning (rC8Q) has largely been improved after the rebuttal

Concerns still outstanding
* evaluation scope (Mkwm, rC8Q) -- the authors added new results on Maze, and acknowledge evaluation under more "realistic" settings as future work, which is reasonable, but the concern could remain.

* novelty (Mkwm) -- this is more subjective. The key claim as I understand from this paper is that, action-centric masked pre-training that targets compositional action structure is the key lever in combinatorial action spaces, and the paper now supports that claim better empirically. Some readers may continue to see the contribution as primarily an effective empirical support of known components rather than a fundamentally new algorithmic principle.

**Reviewer Scores:**

I expect the marginal ratings could bump up, but remains borderline.

---

### Decision · Program_Chairs · 2026-01-26

Accept (Poster)